# B cells inhibit bone formation in rheumatoid arthritis by suppressing osteoblast differentiation

Wen Sun[1,2], Nida Meednu[3], Alexander Rosenberg[3], Javier Rangel-Moreno[3], Victor Wang [3], Jason Glanzman[3], Teresa Owen[3], Xichao Zhou[1], Hengwei Zhang[1], Brendan F. Boyce[1,4], Jennifer H. Anolik[3,4] & Lianping Xing[1,4]

The function of B cells in osteoblast (OB) dysfunction in rheumatoid arthritis (RA) has not been well-studied. Here we show that B cells are enriched in the subchondral and endosteal bone marrow (BM) areas adjacent to osteocalcin[+] OBs in two murine RA models: collagen-induced arthritis and the TNF-transgenic mice. Subchondral BM B cells in RA mice express high levels of OB inhibitors, CCL3 and TNF, and inhibit OB differentiation by activating ERK and NF-κB signaling pathways. The inhibitory effect of RA B cells on OB differentiation is blocked by CCL3 and TNF neutralization, and deletion of CCL3 and TNF in RA B cells completely rescues OB function in vivo, while B cell depletion attenuates bone erosion and OB inhibition in RA mice. Lastly, B cells from RA patients express CCL3 and TNF and inhibit OB differentiation, with these effects ameliorated by CCL3 and TNF neutralization. Thus, B cells inhibit bone formation in RA by producing multiple OB inhibitors.

[1] Department of Pathology and Laboratory Medicine, University of Rochester Medical Center, Rochester, NY 14642, USA. [2] Jiangsu Key Laboratory of Oral Diseases, Nanjing Medical University, 210029 Nanjing, China. [3] Division of Allergy, Immunology and Rheumatology, Department of Medicine, University of Rochester Medical Center, Rochester, NY 14642, USA. [4] Center for Musculoskeletal Research, University of Rochester Medical Center, Rochester, NY 14642, USA. Correspondence and requests for materials should be addressed to J.H.A. (email: jennifer_anolik@urmc.rochester.edu) or to L.X. (email: Lianping_xing@urmc.rochester.edu)

Rheumatoid arthritis (RA) is a chronic inflammatory disease, which affects 1.5 million patients in the United States and causes joint disability in 31% within 4 years of disease onset[1]. Although joint disability in RA can be averted with early aggressive treatment, a major unmet need in the field includes accurately predicting those patients who will accrue progressive joint damage. This requires better definition of the precise immunologic mechanisms of bone loss. Patients with RA often have severe local and systemic bone loss due to increased osteoclast (OC)-mediated bone erosion and decreased osteoblast (OB)-mediated bone formation[2]. Most attention has been focused on the mechanisms responsible for aberrant activation of local joint erosion by OCs, which is mediated by RANKL expressed by several cell types in RA, including synoviocytes[3], B cells[4], and T cells[5]. However, multiple murine models indicate that bone loss in RA is also associated with reduced OB differentiation and bone formation[6,7]. We have demonstrated that OB dysfunction in the TNF transgenic (TNF-Tg) mouse model of RA is mediated by TNF-driven NOTCH activation in mesenchymal precursor cells (MPCs), the precursors of OBs, and similar defects are present in human RA OB precursors[8].

The pathogenesis of RA involves the complex interaction of multiple cell types. B cells play a number of critical roles in RA[9]. They promote auto-immunity through both the production of pathogenic autoantibodies and autoantibody-independent functions, including activation of auto-reactive T cells and production of pro-inflammatory cytokines[2,10–12]. Although B-cell depletion therapy (BCDT) has demonstrated efficacy in a subset of RA patients, the mechanisms by which it ameliorates structural damage in RA are not fully understood. Several studies have indicated that B cells promote OC formation by secreting TNF and RANKL and activating other effector molecules[4,13,14]. However, the effects of B cells in RA on OB differentiation and OB function remain controversial. Studies in TNF-Tg mice[15] and RA patients[16] found that B cells infiltrating the subchondral bone marrow of eroded joints are associated with enhanced bone formation, as evidenced by increased osteoid deposition. In contrast, BCDT in RA patients significantly increases serum levels of procollagen type I amino-terminal propeptide (P1NP), a marker of bone formation, suggesting that the overall effect of B cells on OBs is inhibitory[17]. However, none of these studies has examined the direct effects of B cells on OB differentiation and function.

B cell aggregates in both the synovium and the subchondral bone marrow are well established histopathologic features of RA patients[16]. Within the synovium, B cells can organize into ectopic lymphoid structures and drive T cell activation and propagation as part of the autoimmune response[18]. Further, we have demonstrated recently that B cells within these ectopic structures produce RANKL adjacent to OC precursors and promote osteoclastogenesis in a RANKL-dependent fashion in in vitro cultures[4], suggesting a functional role for B cells in OC-mediated bone erosion in RA. However, the potential influences of B cells on OBs within the target tissue remain unknown, due in part to the challenge of experimental approaches to interrogate these B cells in human tissue.

The current study seeks to further elucidate the mechanisms of immune-mediated joint damage in RA. We use two mouse models of RA, collagen-induced arthritis (CIA) and the TNF-transgenic mice, and demonstrate that in both models B cells are enriched in the subchondral and endosteal bone marrow area, with accumulation close to the bone surface and adjacent to osteocalcin+ OBs. RA B cells from subchondral areas express high levels of several OB inhibitors, including CCL3 and TNF. RA B cells inhibit OB differentiation from MPCs through NF-κB and ERK signaling pathways, which is blocked by CCL3 and TNF neutralization. Deletion of CCL3 and TNF in RA B cells completely abolishes their OB inhibition in vivo. Furthermore, B cells from RA patients express CCL3 and TNF and inhibit OB differentiation, which is blocked by CCL3 and TNF neutralization. Thus, our findings reveal a previously unappreciated role for B cells in RA-associated bone loss and joint damage via direct inhibition of bone formation.

## Results

**B cells are located adjacent to osteoblasts in RA mice**. To investigate the effect of B cells on OBs in RA, we first examined the anatomic relationship between B cells (B220+) and OBs (osteocalcin [OCN]+) in 6-month-old TNF-Tg mice with severe arthritis and systemic bone loss, using double immuno-fluorescence (IF) staining on frozen sections of bone. We examined RA target tissues with a focus on knee synovium, subchondral bone marrow (SBM) of femora and tibiae, and BM of tibiae and patellae. We observed numerous OCN+ OBs on tibial and patellar bone surfaces in samples from WT littermate mice (Fig. 1A). In contrast, OCN+ OB numbers in TNF-Tg mouse samples were markedly reduced, and numerous B cells were detected (Fig. 1A). At low magnification, increased numbers of B cells were seen mainly in SBM areas where they formed aggregates (Fig. 1A and Supplementary Fig. 1A). Flow cytometry analysis of SBM cells confirmed that the B220+ cells are indeed CD19+ B cells (Supplementary Fig. 1B). At high magnification, B cell aggregates were also observed in synovial tissues. B cells in both locations were located close to the bone surfaces, adjacent to OCN+ and ALP+ OBs (Fig. 1B). Occasionally, B cells were observed in cortical defects caused by local bone erosion (Fig. 1B). In long bone BM, increased B cells were located near endosteal surfaces, but they typically did not form aggregates (Fig. 1C). The B220+ B cell area per tissue area (%) was higher in SBM areas than in long bone BM areas (Fig. 1D), while the subchondral bone volume was significantly reduced in TNF-Tg mice compared to their WT littermates (Fig. 1E). The demonstration of numerous Ki67+ B cells in the SBM areas of TNF-Tg mice suggests that B cells are locally activated and proliferating (Supplementary Fig. 2). Based on IF staining with anti-B220 Ab, the overall B cell distribution in long bones and joints of a TNF-Tg mouse is illustrated pictorially (Supplementary Fig. 3).

To further demonstrate the close relationship between B cells and OBs in RA, we examined B cells and OBs in CIA mice[19]. Arthritis is induced in this model via the immunization of genetically susceptible strains of mice with type II collagen in an adjuvant and is B cell-dependent[20]. Mice develop arthritis and systemic bone loss after the onset of joint lesions[21]. Similar to TNF-Tg mice, we detected numerous B220+ B cells that were located adjacent to OCN+ OBs in SBM and endosteal BM areas of CIA mice (Fig. 1F). This B cell expansion in the SBM was associated with a significant reduction in OCN+ OB area per tissue area in CIA compared to control mice (Fig. 1G&H).

**RA B cells express high levels of osteoblast inhibitors**. Given the close anatomic relationship between B cells and OBs in TNF-Tg and CIA mice joints, we hypothesized that RA B cells especially in the SBM area may produce factors that affect OB function. We first characterized B cells by flow cytometry to determine if B cell development is altered in the RA mice. B cell distribution into developmental subsets is not different in the BM and blood of the TNF-Tg vs. WT mice (Supplementary Fig. 4). However, the SBM B cells from TNF-Tg mice are phenotypically distinct from total BM B cells (Supplementary Fig. 4 and Supplementary Fig. 5). In particular, the SBM is enriched in IgM+IgD+AA4.1- mature B cells (Supplementary Fig. 5).

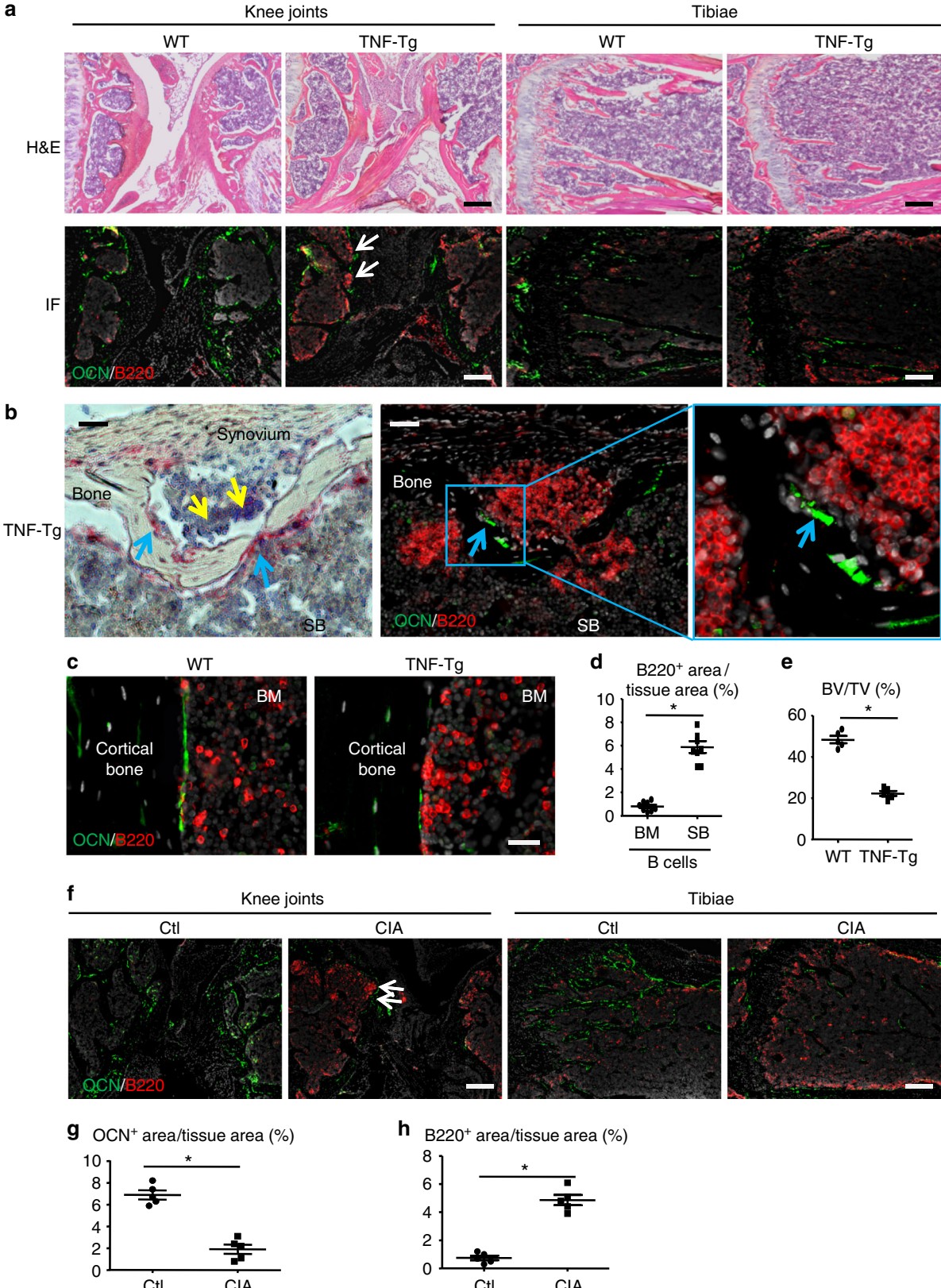

We purified CD19+ B cells from BM of TNF-Tg mice, BM of WT littermates, and SBM (including synovium) of TNF-Tg mice and performed RNA-seq to compare the transcriptome profiles in these 3 groups (Fig. 2A). Using pooled differential expression analysis, we identified more differentially expressed genes in TNF-Tg SBM B cells vs. TNF-Tg BM B cells than TNF-Tg BM B cells vs. WT BM B cells (464 vs. 65 at $q < 0.001$, Fig. 2B). At a less restrictive cutoff ($p < 0.05$), there were 99 genes in common between the two comparisons. Interestingly, most (77/90) of the common, up-regulated genes in the TNF-Tg comparison (TNF-

**Fig. 1** B cells are located close to osteoblasts in bones from RA mice. **a** Frozen sections of knees from 6-m-old TNF-Tg and WT mice were H&E stained (upper panels) and adjacent sections (lower panels) were subjected to IF with anti-B220 Ab for B cells (red) and anti-osteocalcin (OCN) Ab for OBs (green). White arrows = B cell aggregates. Bar = 200 μm. **b** ALP staining (left-hand panel) and an adjacent section (middle panel and right panel at higher magnification) stained for B cells and OBs. Yellow arrows: inflammatory cells. Blue arrows: OBs. Bar = 50 μm. **c** Images of long bones from the above mice in **a** show B cells and OBs along the endosteum of cortical bone of a TNF-Tg mouse. Bar = 25 μm. **d** B220+ B cell area/tissue area (%) in long bone marrow (BM) and SB areas from TNF-Tg mice were measured on B220 IF-stained sections. *p < 0.05 vs. BM. **e** μCT analysis and morphometric data of femoral SB volume. *p < 0.05 vs. WT. **f** Frozen sections of knees from Ctl and CIA mice 2 weeks after the onset of CIA were subjected to IF with anti-B220 and anti-OCN Abs. White arrows = B cell aggregate. Bar = 200 μm. The quantification of OCN+ area/tissue area (%) (**g**) and B220+ B cell area/tissue area (%) (**h**) in SB area from the above mice in **f**. *p < 0.05 vs. Ctl. 5–8 mice and their controls were included in each experiment. All of these experiments were repeated at least once. Representative images and quantifications in the panels come from one independent experiment. All error bars represent s.e.m. Two-tailed unpaired Student's t-test was performed

Tg SBM B cells vs. TNF-Tg BM B cells) were also up-regulated in the BM B cell comparison (TNF-Tg BM B cells vs. WT BM B cells). The 30 most differentially expressed genes for each comparison (Supplementary Table 1 and Supplementary Table 2) were submitted to pathway analysis (Supplementary Fig. 6) and included genes associated with OC differentiation, RA, TNF signaling, and chemokine signaling pathways. Of note, the expression levels of several genes encoding putative OB inhibitors, including CCL3 (also known as macrophage inflammatory protein 1-alpha, MIP-1-alpha), TNF, and Dkk3, were much higher in TNF-Tg SBM B cells than in their BM counterparts, which we confirmed by qPCR in an independent cohort of mice (Fig. 2C). In marked contrast, OB inhibitor expression levels were similar between WT SBM B cells and their BM counterparts (Supplementary Fig. 7).

To confirm the data from TNF-Tg mice, we isolated B cells from BM of CIA mice and Control (Ctl) mice and B cells from the SBM (including synovium) of CIA mice to compare mRNA expression levels of the OB inhibitors, CCL3, TNF and Dkk3. Levels of all three inhibitors were markedly higher in CIA SBM B cells than in their BM counterparts, while no difference was detected in BM B cells between CIA and Ctl mice (Fig. 2D). These data suggest that B cells in RA target tissues, especially in the SBM, may inhibit OBs.

**RA B cells inhibit osteoblasts via NF-κB and ERK signaling.** We next investigated if B cells can directly inhibit OB differentiation. As an initial approach, conditioned medium from TNF-Tg or WT BM B cell was cultured with WT mesenchymal precursor cells (MPCs) in OB-inducing medium. B cells were first stimulated with anti-CD40Ab + IL4 + LPS, conditions that promote TNF production by mouse B cells (Supplementary Fig. 8A). B cell conditioned medium inhibited OB differentiation in a dose-dependent manner, as measured by reduced ALP+ staining areas. TNF-Tg B cells were more potent inhibitors than control B cells (Fig. 3A). As a control, the medium used to stimulate the B cells had no effects on OB differentiation (Supplementary Fig. 8B). Next, we directly co-cultured B cells with WT MPCs in OB-inducing medium. Compared to WT B cells, TNF-Tg B cells significantly reduced OB differentiation, as seen by reduced ALP+ staining areas (Fig. 3B) and decreased expression of the OB-associated genes, *Runx2* and *ALP* (Fig. 3C). To further confirm the data from TNF-Tg mice, we directly co-cultured SBM or BM B cells from CIA mice with WT MPCs in OB-inducing medium. SBM B cells of CIA mice significantly reduced OB differentiation of MPCs, as seen by reduced ALP+ staining areas (Fig. 3D) and decreased expression of the OB-associated genes, *Runx2* and *ALP* (Fig. 3E). However, we did not observe any difference in the effects of BM B cells from CIA and Ctl mice on OBs.

Given the markedly increased levels of CCL3, TNF, and Dkk3 in RA SBM B cells, all of which can inhibit OBs[22–24], we next

examined for changes of signaling pathways in MPCs downstream of these factors (CCL3 activates ERK signaling[25], TNF activates NF-κB signaling[26], and Dkk3 inhibits β-catenin signaling[27]). By Western blot analysis, the expression levels of NF-κB proteins, p-AKT, and p-ERK were increased in MPCs, while expression of β-catenin signaling proteins was unchanged in MPCs when they were co-cultured with TNF-Tg B cells (Fig. 3F). To further test if B cell-produced CCL3 and TNF mediate the inhibitory effect on OBs, we treated B cell-MPC co-cultures with a CCL3 and/or TNF neutralizing Abs in OB-inducing medium. CCL3 or TNF neutralization partially rescued the inhibitory effect of TNF-Tg B cells on MPCs (Fig. 4A) and the NF-κB (Fig. 4B) and ERK (Fig. 4C) activation in these cells. Further, neutralization of both CCL3 and TNF completely rescued the B cell inhibitory effect (Fig. 4A-C).

**RA B cells inhibit osteoblast function via CCL3 and TNF.** CCL3 is a newly identified inhibitor of OBs in multiple myeloma[22,25] and leukemia[28]. We confirmed that CCL3 significantly reduced OB differentiation from MPCs in a dose-dependent manner (Fig. 5A), and it also inhibited expression of the OB-associated genes, *Runx2* and *ALP* (Fig. 5B). Further, B cells in the SBM of TNF-Tg mice expressed high levels of CCL3 by IF staining (Fig. 5C). ELISA analysis revealed that B cells produce more CCL3 when stimulated, and B cells from TNF-Tg mice produced higher level of CCL3 than B cells from WT mice (Fig. 5D).

Several studies have demonstrated that TNF can be produced by B cells[29] and promote OC and other effector cell activation[14,30]. However, the effects of B cell-produced TNF on OB inhibition in RA have not been studied. We first demonstrated that B cells from WT and TNF-Tg mice produced more TNF in culture after stimulation, and B cells from TNF-Tg mice produced higher levels of TNF than B cells from WT mice (Fig. 5E). As a control, B cells from TNF knockout mice (TNF-KO) did not produce TNF (Supplementary Fig. 8C) and, similar to the effects of the TNF neutralizing Ab on B cells, they had a lower inhibitory effect on OB differentiation than B cells from WT and TNF-Tg B (Supplementary Fig. 8D).

To further examine the inhibitory effect of RA B cells on osteoblastic bone formation in vivo, we mixed WT MPCs with BM B cells from WT, TNF-Tg, TNF-Tg/CCL3-KO, TNF-Tg/TNF-KO, or TNF-Tg/CCL3-KO/TNF-KO mice in Gelfoam and surgically implanted them subcutaneously into recipient SCID mice in a standard bone formation assay[31]. As expected, the volume of bone formed when TNF-Tg BM B cells were co-implanted with WT MPCs was significantly lower than that formed in response to WT BM B cells (Fig. 5F-H). In marked contrast, the volumes of bone formed in response to B cells from TNF-Tg/CCL3-KO BM and TNF-Tg/TNF-KO BM were higher than that formed in response to TNF-Tg BM B cells (Fig. 5F-H). Importantly, the B cell inhibitory phenotype was completely

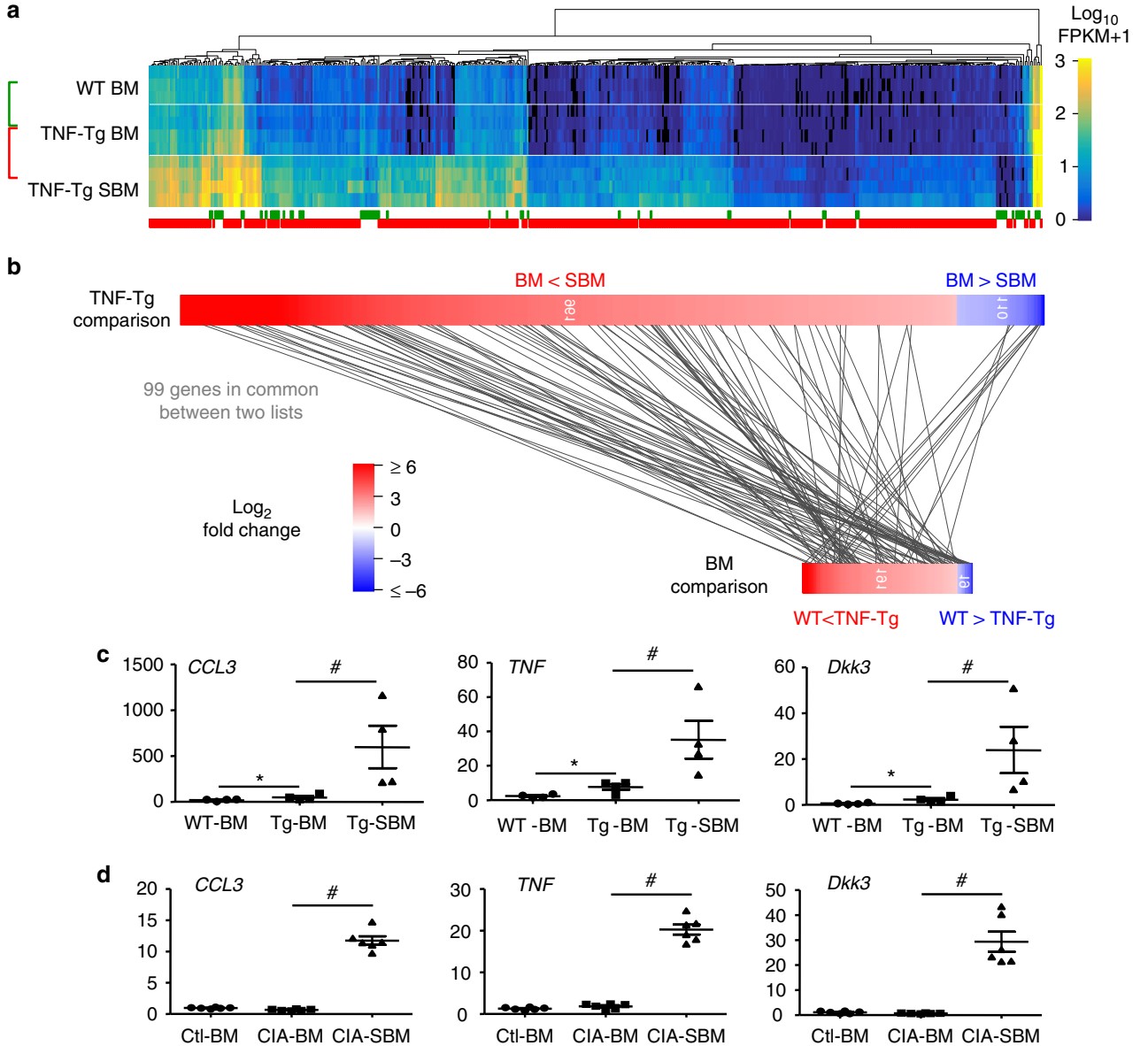

**Fig. 2** B cells from RA express high levels of osteoblast inhibitors. B cells were purified from 6-m-old WT BM, TNF-Tg BM, and TNF-Tg subchondral BM (TNF-Tg SBM) with anti-CD19 beads, $N = 3–4$ mice per group. **a** Using pooled differential expression analysis, genes were found where $q = <0.001$ for either the BM comparison (WT vs. TNF-Tg, $N = 65$, green), or the TNF-Tg comparison (BM vs. SBM, $N = 464$, red). Identified genes where $q = <0.05$ for any within-group, between-replicate pairwise comparisons were omitted, resulting in 508 differentially-expressed genes. Colors correspond to log10 FPKM+ 1. **b** Comparison of significant gene lists for the TNF-Tg (upper) comparison (BM vs. SBM) and BM (lower) comparison (WT vs. TNF-Tg). Gene lists constructed similar to **a** except $q < 0.05$ for the pooled group comparisons. Each horizontal bar represents the significant genes ordered by fold-change (color). Numbers in the vertical bar indicate fold-change in gene expression levels. Genes present in both lists ($n = 99$) are indicated by connecting gray lines. **c** The gene expression levels were measured by qPCR, $N = 4$ mice per group. Values were calculated based the equation= ½ CT (gene of interest) – CT (*Gapdh*) ×100. *$p < 0.05$ vs. WT-BM, #$p < 0.05$ vs. Tg-BM. **d** B cells were isolated from Ctl BM, CIA BM and CIA SBM with anti-CD19 beads, 2 weeks after CIA onset. The gene expression levels were measured by qPCR, $N = 6$. #$p < 0.05$ vs. CIA-BM. **a** and **b**) were performed once with 3–4 mice in each group. 4–6 mice from an independent cohort were included in **c** and **d** with the experiment repeated at least once. Representative pictures and quantifications are shown. All error bars represent s.e.m. One way ANOVA followed by Dunnett's post-hoc multiple comparisons was performed for **c** and **d**

abrogated by implanting B cells from TNF-Tg/CCL3-KO/TNF-KO mice. We confirmed that B cells survive surgical implantation by implanting MPCs from GFP transgenic mice along with BM B cells from mTmG mice (Fig. 5I).

**BCDT increases numbers of osteoblasts in TNF-Tg mice**. To further investigate the importance of B cells in OB inhibition in vivo, we treated TNF-Tg mice with murine anti-CD20 Ab once

a week for 8 weeks. This treatment regimen has been shown previously to fully deplete mature B cells from the spleen and the BM without affecting BM B cell precursors[32]. Therefore, it allowed us to examine the role of mature B cells in osteo-blastogenesis. We found that BCDT increased body weight and decreased spleen size in TNF-Tg mice (Fig. 6A). Analysis of cell populations by flow cytometry revealed a significant reduction of B220+ B cells in the spleen after depletion (Supplementary Fig. 9A). Importantly, BCDT markedly reduced B cell aggregates

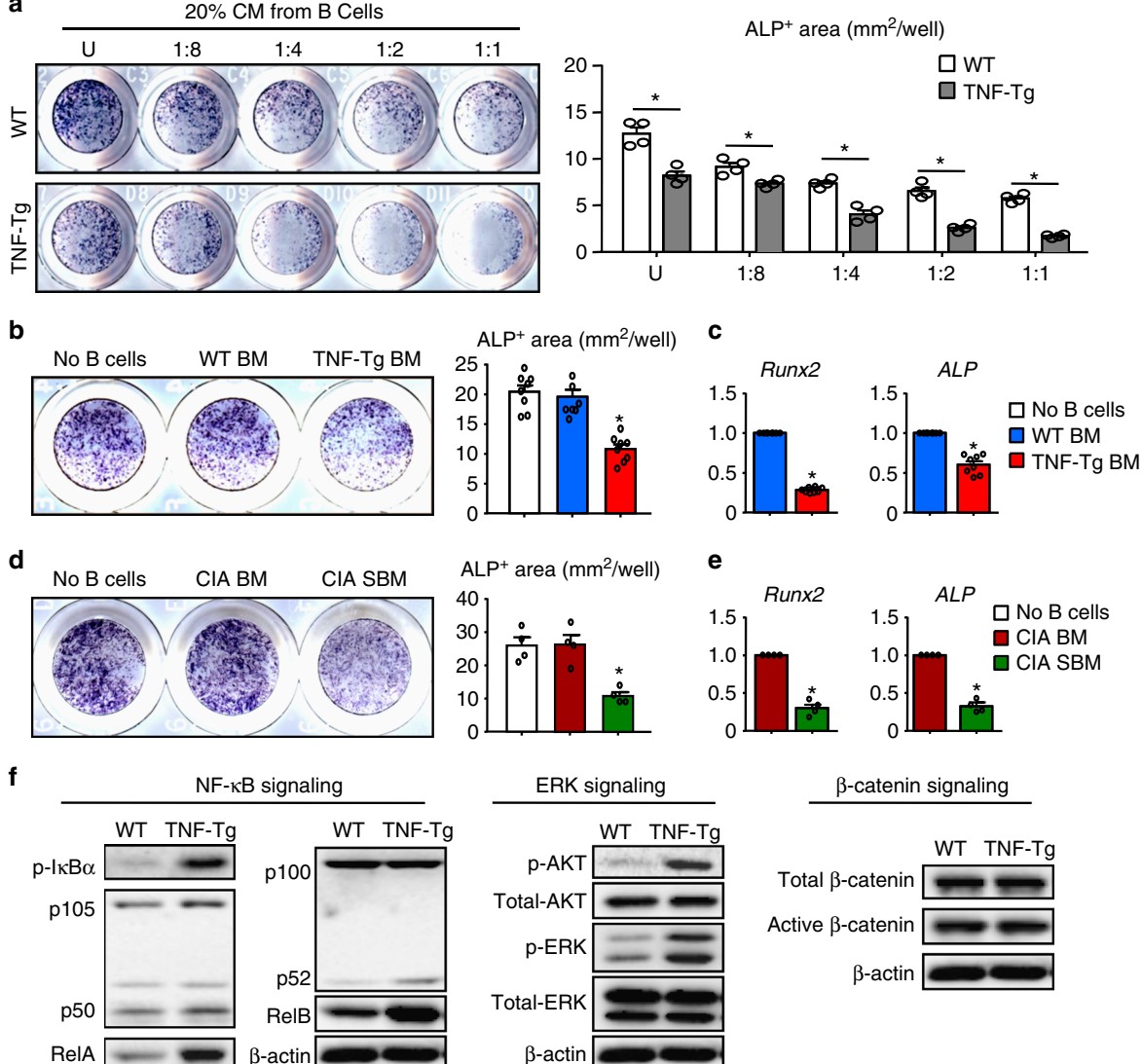

**Fig. 3** RA B cells inhibit osteoblast differentiation and activate NF-κB and ERK signaling. **a** B cells were purified from 6-m-old WT and TNF-Tg mouse BM, as in Fig. 2A. Mesenchymal precursor cells (MPCs) were cultured for 2 days in OB-inducing medium + 20% CM from untreated (U) B cells or from cultures of previously stimulated B cells. These B cells had been treated for 24 h with 1/8, 1/4, 1/2, and 1/1 dilutions of the optimal concentrations of the stimulatory cocktail (2.5 μg/ml Anti-CD40+10 ng/ml IL4+10 μg/ml LPS). Culture plates were stained for ALP, and ALP+ areas were measured. *$p < 0.05$ vs. WT. **b** WT MPCs were cultured ± B cells in OB-inducing medium for 2 days. The area of ALP+ cells was measured. *$p < 0.05$ vs. WT. **c** Expression levels of *Runx2* and *ALP* in cells from **b** were measured by qPCR. *$p < 0.05$ vs. WT. **d** B cells were isolated from BM or subchondral BM (SBM) of CIA mice and co-cultured with WT MPCs in OB-inducing medium for 2 days. The area of ALP+ cells was measured. *$p < 0.05$ vs. BM. **e** Expression levels of *Runx2* and *ALP* in cells from **d** were measured by qPCR. *$p < 0.05$ vs. BM. **f** WT and TNF-Tg B cells were co-cultured with WT MPCs for 2 days. At the end of the culture period, B cells were removed and protein lysates from the MPCs were subjected to Western blot analyses. Expression of NF-κB, ERK, and β-catenin signaling molecules in cell lysates from MPCs were assessed. Supplementary Fig. 13 shows uncropped gel images. Each experiment was performed 3 to 8 times. Representative images and quantifications are shown from one independent experiment. All error bars represent s.e.m. Two-tailed unpaired Student's *t*-test was performed

in SBM areas, which was accompanied by a marked increase in the area of OCN+ OBs (Fig. 6B).

To determine if BCDT attenuates bone loss of TNF-Tg mice, we focused on knee joints because B cell aggregates are enriched in this location (Fig. 1A&B) and are reduced by BCDT in the subchondral bone area around knees (Fig. 6B). Although early studies reported that TNF-Tg/RAG1$^{−/−}$ mice (lacking mature T or B cells) still develop erosive arthritis in ankle joints[33], μCT revealed that BCDT significantly increased patellar and subchondral bone volume in TNF-Tg mice (Fig. 6C). Increased bone volume was confirmed in BCDT-treated mice by histomorphometric analysis in H&E-stained sections (Fig. 6D) and this was

associated with increased OB numbers (Fig. 6E). Because B cells produce RANKL in RA patients to stimulate osteoclastogenesis and induce bone loss[4], we also examined OC numbers in adjacent TRAP-stained sections and found that OC numbers were decreased in subchondral bone areas in mice given BCDT (Fig. 6F). BCDT also decreased the area of inflammation and bone erosion in the knees (Supplementary Fig. 9B&C) and ankles of TNF-Tg mice (Supplementary Fig. 9D&E).

To determine if BCDT affects OB differentiation, we examined CFU-F and CFU-ALP+ colony and nodule formation using BM stromal cells from BCDT- or IgG-treated TNF-Tg mice. Cells from B cell-depleted mice formed significantly increased numbers

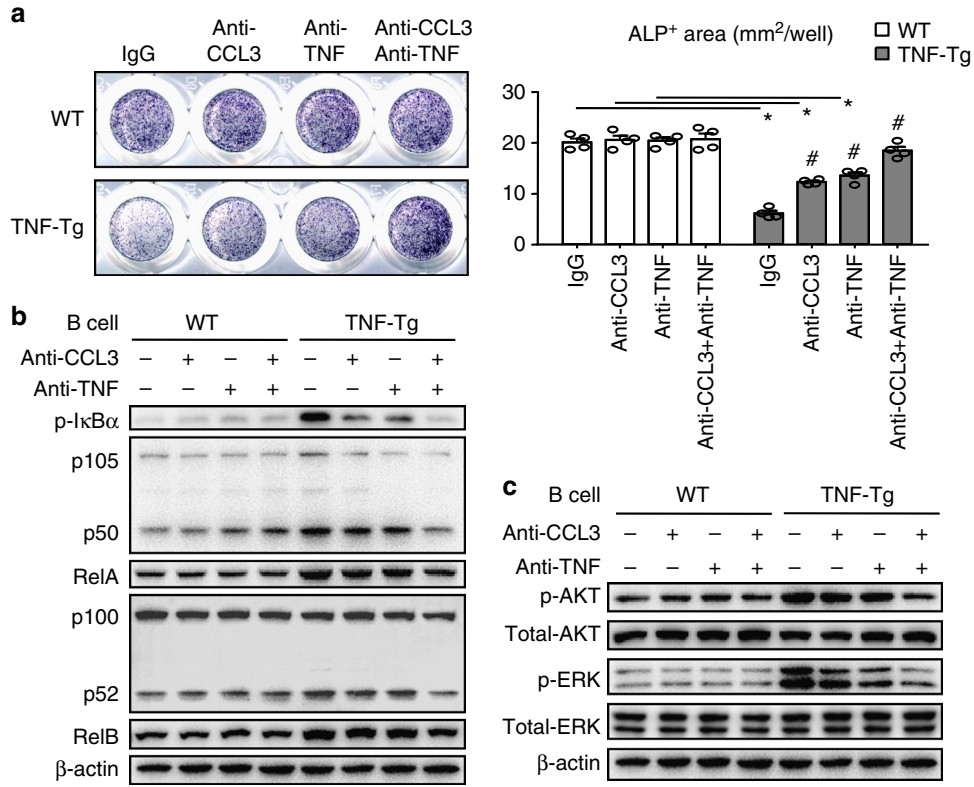

**Fig. 4** B cells inhibit osteoblast differentiation in vitro by secreting CCL3 and TNF. **a–c** BM B cells from WT or TNF-Tg mice were purified, as in Fig. 2A. B cells were co-cultured with MPCs ± anti-CCL3 neutralizing Ab ± anti-TNF neutralizing Ab in OB-inducing medium for 2 days. **a** ALP+ area was measured. *$p < 0.05$ vs. WT, #$p < 0.05$ vs. TNF-Tg treated with IgG. (**b**, **c**) B cells were removed and protein lysates from the MPCs were subjected to Western blot analyses. Expression levels of NF-κB (**b**) and ERK signaling (**c**) molecules in cell lysates from MPCs were assessed. Supplementary Fig. 14 shows uncropped gel images. Each experiment was performed 3 to 5 times. Representative images and quantifications shown in the figure come from one independent experiment. All error bars represent s.e.m. One way ANOVA followed by Dunnett's post-hoc multiple comparisons was performed

of CFU-F and CFU-ALP+ colonies and nodules compared to cells from control mice (Fig. 6G). Additionally, BM cells from mice given BCDT formed fewer OCs than cells from control mice (Fig. 6H). Thus, BCDT attenuates bone loss and erosion in TNF-Tg mice by affecting both OC and OB lineage cells.

**B cells from RA patients inhibit osteoblast differentiation**. To determine the clinical relevance of our mouse findings, we examined the effect of stimulated B cells from normal subjects and B cells isolated from RA patients on differentiation of normal human mesenchymal stem cells (hMSCs) into OBs. Purified B cells from peripheral blood (PB) of healthy controls produced increased amounts of CCL3 and TNF in response to treatment with CpG2006 + anti-Ig (A+G+M) (Fig. 7A&B). Conditioned medium from stimulated B cells had a similar inhibitory effect on OB formation as mouse RA B cells, which was partly prevented by CCL3 or TNF neutralizing Abs and was completely abrogated by CCL3 and TNF double blockade (Fig. 7C).

Next, B cells were isolated from PB of healthy controls or RA patients with active disease and co-cultured with hMSCs in OB-inducing medium. RA B cells significantly inhibited OB differentiation, associated with reduced ALP+ staining area (Fig. 7D) and expression of *Runx2* and *ALP* (Fig. 7E). Finally, examination of synovial samples from RA patients revealed aggregates of CD20+ B cells, which expressed CCL3 and TNF (Fig. 7F), demonstrating that B cells in human RA target tissue indeed express OB inhibitory factors. Furthermore, our pre-liminary findings revealed that memory B cells, which we previously demonstrated are enriched in the RA synovium[4], but

not plasma cells, express the highest levels of OB inhibitors and mediate the greatest inhibition of OB differentiation (Supplementary Fig. 10).

## Discussion

B cells play critical roles in the pathogenesis of RA via both Ab-dependent and Ab-independent mechanisms. However, the contribution of B cells to bone loss and erosion remains unclear. In this study, we have demonstrated for the first time that RA B cells directly inhibit mesenchymal precursor cell differentiation into bone-forming OBs by producing an array of OB inhibitory factors, including CCL3 and TNF. BCDT prevented bone loss in RA mice, associated with increased numbers of OBs. We found that B cell-induced OB inhibition and bone loss were most pro-nounced in the subchondral bone area, directly adjacent to inflamed RA synovium. Furthermore, B cell-derived CCL3 and TNF directly mediate bone loss in vivo, given that deletion of these factors in B cells restored OB activity in an ectopic bone formation model. Importantly, we confirmed these findings in RA patients by demonstrating that B cells in peripheral blood and synovial tissue express CCL3 and TNF and also inhibit OB dif-ferentiation, which is prevented by CCL3 and TNF neutralization. Thus, our studies have revealed a new role for B cells in RA-induced bone loss and erosion by directly inhibiting OB differ-entiation via an Ab-independent mechanism.

B cells have recently been appreciated as contributors to bone loss in RA, independent of Ab production, by stimulating OC-mediated bone resorption through expression of RANKL and TNF[4,30]. However, the relationship between B cells and OB

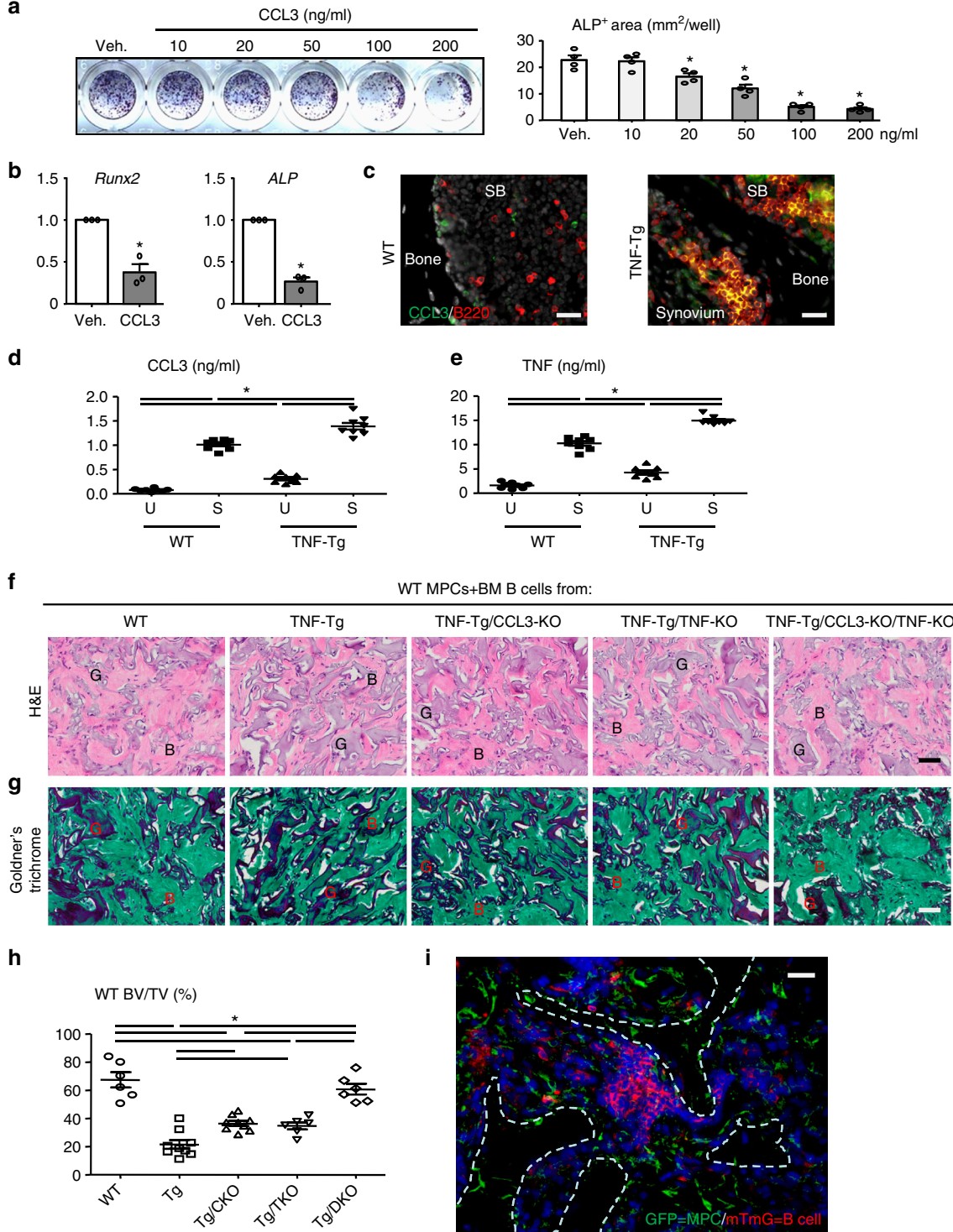

lineage cells in RA has not been well studied. An early study reported that aggregates of B cells in the subchondral BM of TNF-Tg mice express BMP-7 and these B cells are located close to OBs in affected joints at sites of bone erosion[15]. The authors speculated that OBs are activated by B cell-derived BMP7 in an attempt to repair damaged bone, and suggested that B cells may stimulate bone formation in RA. Along these lines, chemokines, such as CXCL13 (a B lymphocyte chemoattractant), that are produced in ectopic lymphoid aggregates have been suggested to provide additional signals for recruitment and activation of OBs, as well as B cells[34,35]. However, our findings indicate a clear inhibitory

role for B cells on OB differentiation: B cells from both RA mouse models (CIA and TNF-Tg mice) and RA patients exerted an inhibitory effect when they were co-cultured with mesenchymal precursor cells, and BCDT markedly increased OB numbers and bone volume in TNF-Tg mice. This supports our hypothesis that the dominant effect of B cells in RA on bone formation is inhibitory. It is important to note that BCDT targets predominantly CD20-expressing mature B cells, which may function in an OB inhibitory role. Distinct roles for different B cell subsets may also explain why more global B cell-deficient mice, such as Pax5 mutants, µMT, and Rag1 mice do not have an osteoblast

**Fig. 5** RA B cells express high levels of CCL3 and TNF and inhibit osteoblast bone formation in vivo. **a** MPCs were treated with different doses of CCL3 in OB-inducing medium for 2 days. The area of ALP+ cells was measured. *$p < 0.05$ vs. Veh. **b** Expression levels of *Runx2* and *ALP* in 50 ng/ml CCL3 treated cells from **a** were measured by qPCR. Fold-changes were calculated by dividing the values with those from Veh. *$p < 0.05$ vs. Veh. **c** CCL3+ B cells (yellow) were detected in subchondral bone (SB) area in frozen sections of proximal tibiae from TNF-Tg mice and their WT littermates by double IF staining with anti-B220 (red) and anti-CCL3 (green) Abs. Bar = 25 μm. **d, e** BM B cells from WT or TNF-Tg mice were purified and stimulated (S) with 2.5 μg/ml anti-CD40 Ab +10 ng/ml IL4 +10 μg/ml LPS or vehicle (U) for 4 h. CCL3 (**d**) and TNF (**e**) protein expression levels were assessed in the culture media by ELISA. **f–h** WT MPCs were transplanted with BM B cells from WT, TNF-Tg, TNF-Tg/CCL3-KO, TNF-Tg/TNF-KO, and TNF-Tg/CCL3-KO/TNF-KO mice by subcutaneous surgical implantation into recipient SCID mice. 4 weeks later, the implants were harvested and H&E staining (**f**) and Goldner's Trichrome (**g**) staining were performed. B bone. G GelFoam. Bar = 50 μm. **h** A histomorphometric analysis of bone volume to tissue volume in H&E-stained sections. **i** MPCs from GFP transgenic mice were transplanted with BM B cells from mTmG mice. Representative images show mTmG+ B cells (red) and GFP+ MPCs (green) in the implants. White dashed lines indicate the bone surface. Bar = 25 μm. *$p < 0.05$ as indicated groups. Each experiment was performed 3 to 5 times. Representative images and quantifications are shown from one independent experiment. All error bars represent s.e.m. Two-tailed unpaired Student's *t*-test was performed for **b**. One way ANOVA followed by Dunnett's post-hoc multiple comparisons was performed for all the others

phenotype[36,37]. These observations may be attributable to the simultaneous absence of B cells that negatively and positively impact OB function. Furthermore, our preliminary findings revealed memory B cells in humans as the cell subset that expresses high levels of OB inhibitors and maximally inhibits OB differentiation (Supplementary Fig. 10). Notably, in human RA memory B cells also dominate the synovial B cell infiltrates[4].

More study is needed to define how the B cell OB inhibitory effect may change with B cell development and activation within the inflammatory joint milieu. Based on our transcriptome analysis and the presence of numerous Ki67+ B cells in situ, we speculate that local B cell activation contributes to the OB inhibitory phenotype in murine synovium and subchondral BM, as well as in human RA. Given prior publications demonstrating plasma cell infiltrates in the RA synovium[38] and migration of plasma cells to inflamed tissue[39,40], it is important to note that the subchondral bone marrow/synovium CD19+ B cells do not have a plasma cell phenotype and transcriptome analysis does not demonstrate a typical plasma cell signature, e.g., enrichment for genes like XBP1 or BLIMP1 or pathways, such as the unfolded protein response and intracellular protein transport[41]. However, it remains possible that some pathogenic B cells are generated in other sites, such as the secondary lymphatics, and migrate to find a survival niche in the joint target tissue. Regardless of the relative contribution of joint in situ generation vs. migration, it is clear that the B cells in the target tissue have a distinct phenotype and function that significantly impacts bone homeostasis.

We found that RA B cells produce much higher levels of CCL3, which inhibits MPCs differentiation to OBs. CCL3 is a C–C family chemokine detected at high levels in the synovial tissue and fluids of RA patients[42]. An early study reported that naïve, memory, and germinal center B cells all produced CCL3 in response to BCR triggering using human tonsillar B cells[43]. A genome-wide study identified CCL3 as a risk locus for RA[44]. CCL3 null mice were resistant to the development of inflammation and joint destruction induced by anti-type II collagen monoclonal antibody[45]. In addition, wild-type and CCL3 null mice produced comparable levels of TNF in an LPS challenge experiment[45], suggesting that CCL3, but not TNF, plays a predominant role in the pathogenesis of anti-type II collagen monoclonal antibody-induced RA. Apart from B cells, many cell types, including macrophages and T cells, express CCL3. We compared CCL3 expression levels in B cells, T cells, macrophages, and CD45-mesenchymal lineage cells and found that activated B cells from normal subjects produce similar levels of CCL3 as macrophages (Supplementary Fig. 11). Further, the requirement of B cell-produced CCL3 in OB inhibition was directly examined by transplanting BM B cells with MPCs by surgical implantation into recipient mice. Notably, B cell deletion of CCL3 partially rescued, while knock-out of both CCL3 and TNF in the B cell

compartment completely rescued the bone loss caused by TNF-Tg BM B cells. However, the requirement of B cell-produced CCL3 in RA pathogenesis and bone loss merits further examination in mice carrying the B cell-specific knockout of CCL3 or OB-specific or mesenchymal precursor-specific knockout of CCR1/CCR5, receptors for CCL3.

Our RNA-sequencing results indicate that in addition to CCL3, B cells also produce high levels of other OB inhibitory factors, such as TNF and Dkk3, which are expressed in RA synovium[7,46]. This raises two important points. First, in RA B cells produce multiple OB inhibitors which likely have overlapping functions. We speculate that it may not be sufficient to inhibit a single factor in B cells to prevent OB inhibition in RA. Identification and characterization of specific B cell subsets in RA target tissues, such as the subchondral bone area, that produce high levels of OB inhibitors may allow more targeted therapeutic approaches to prevent bone loss. A better understanding of the micro-environmental signals in RA target tissues that regulate B cell production of these OB inhibitors and the relative contribution of each to OB dysfunction and erosion progression in RA could also lead to new targets for therapeutic intervention. Secondly, our study clearly demonstrated involvement of B cells in both the inflammatory arthritis and the inflammatory bone loss in TNF-Tg mice. This contrasts with some published literature suggesting that B cells do not play a critical role in the TNF-Tg mouse model because TNF-Tg/Rag1$^{-/-}$ mice still develop arthritis[33]. This discrepancy may reflect the use of different TNF-Tg mouse lines. For example, the 197 TNF-Tg mouse line that was used to generate TNF-Tg/Rag1$^{-/-}$ mice carries multiple copies of the TNF transgene, with a very aggressive arthritis at a young age. In contrast, the 3647 line of TNF-Tg mice that we used carries one copy of the TNF transgene and develops paw and knee arthritis at a much slower pace[47]. Furthermore, we confirmed that B cells from mice with CIA have a similar OB inhibitory mechanism as TNF-Tg B cells, indicating that B cell-secreted factors are likely involved in the negative regulation of OB differentiation in RA.

In summary, using TNF-Tg and CIA mouse models and human RA samples, we have demonstrated for the first time that B cells inhibit OB differentiation in RA by producing an array of negative regulators of OBs within the bone marrow micro-environment contributing to bone loss and erosion in RA. Our findings define a novel role for B cells in the pathogenesis of bone loss in RA and support the development of therapies targeting specific pathogenic B cells and key inhibitors of OB differentiation factors contributing to joint damage.

## Methods
**Animals.** (1) A line of TNF-transgenic (TNF-Tg) mice (line 3647) carrying a modified human TNF transgene in which the 3′-region of the TNF gene was replaced with that of the human α-globin gene. We have demonstrated that

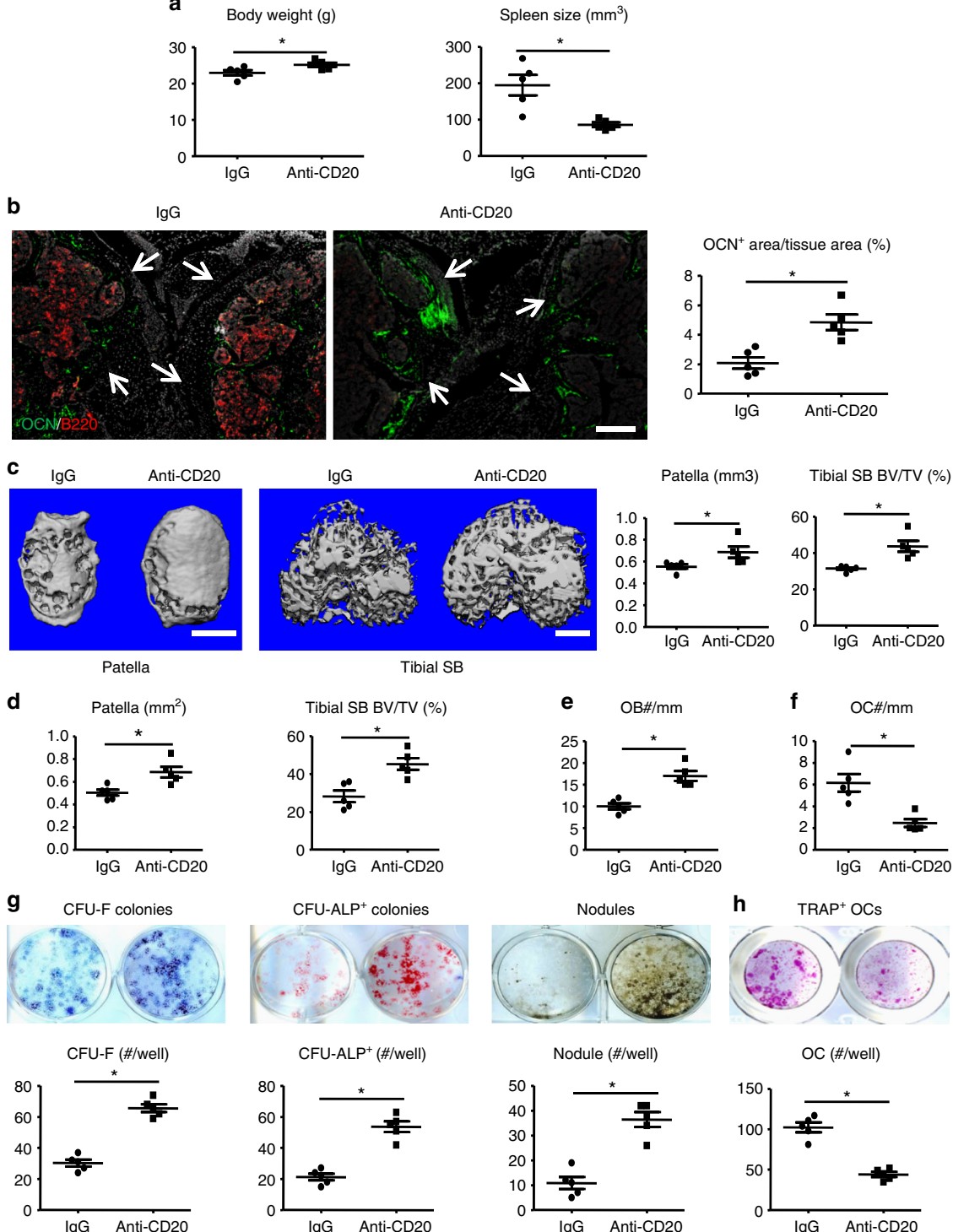

**Fig. 6** B cell depletion therapy increases osteoblast numbers in TNF-Tg mice. 2.5-month-old TNF-Tg mice and WT littermates ($N = 5$/group) were given murine anti-CD20 Ab (10 mg/kg/injection intravenously) or isotype IgG once a week for 8 weeks for B cell depletion therapy (BCDT). **a** Effects of BCDT on body weight and spleen size. **b** Double IF staining of frozen sections of knees from TNF-Tg mice with anti-B220 (red) and anti-OCN (green) Abs showing a reduction in B cells in the SB area in BCDT groups (middle panel) compared with IgG groups. Arrows indicate the location of articular cartilage. Bar = 200 μm. The right-hand panel shows quantification of OCN+ area/tissue area (%) in these mice. **c** Representative μCT scans and bone volumes in patellae and tibial SB areas. Bar = 500 μm. **d** Histomorphometric analysis of patellar and tibial SB volume. **e** OB numbers per mm bone surface in H&E-stained sections of tibial SB areas. **f** OC numbers per mm bone surface in TRAP-stained sections of tibial SB areas. **g** BM cells were cultured in basal or OB-inducing medium for 12 days in CFU colony or 21 days in nodule formation assays. The numbers of CFU-F, CFU-ALP+ colonies and nodules were evaluated. **h** BM cells were cultured with RANKL and M-CSF for 5 days. TRAP+ OC numbers were counted. *$p < 0.05$ vs. IgG-treated mice. 5 mice and their controls were included in each experiment. All of these experiments were repeated once (representative images and quantifications shown). All error bars represent s.e.m. Two-tailed unpaired Student's $t$-test was performed

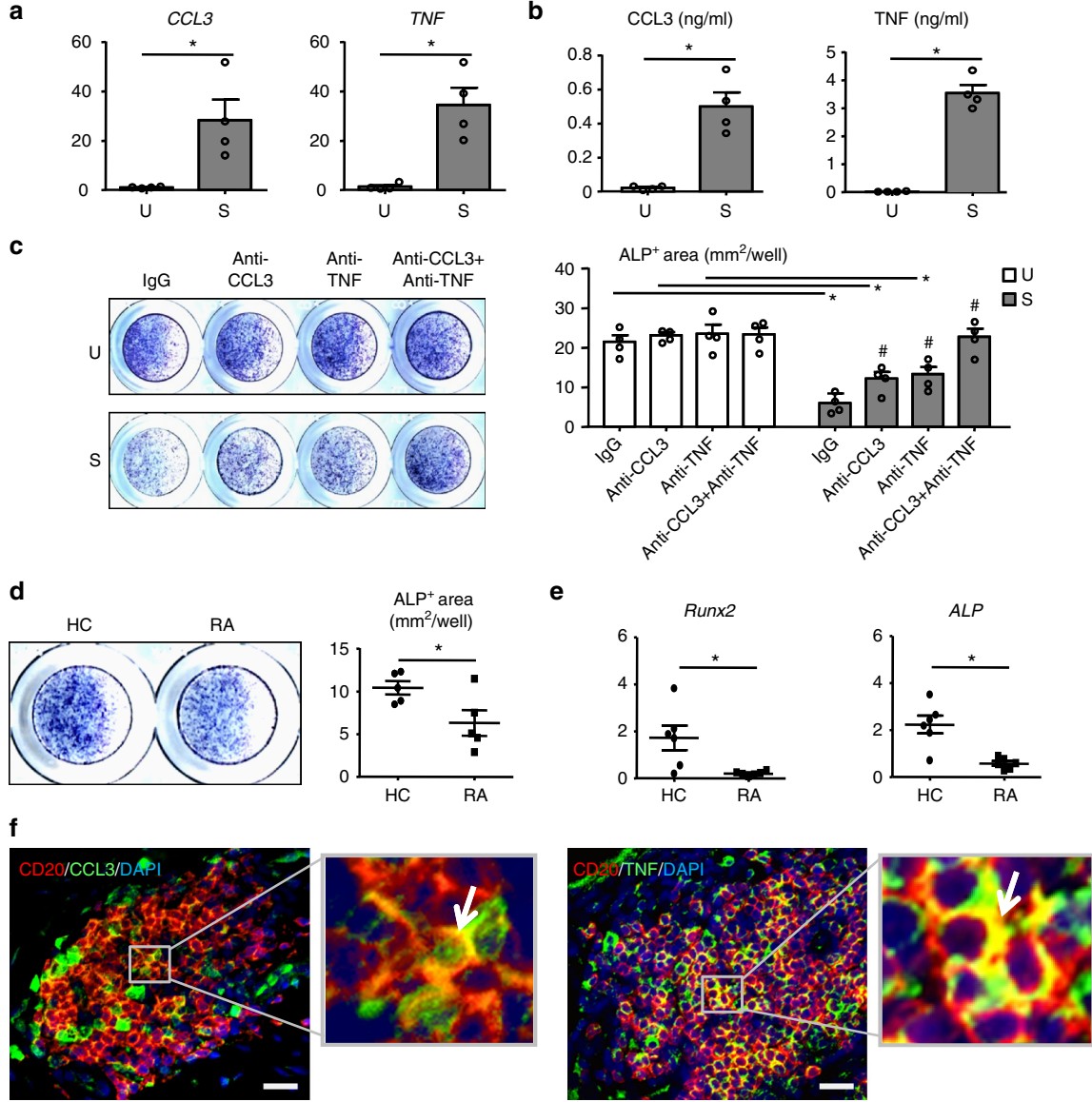

**Fig. 7** B cells from RA patients inhibit OB differentiation. **a–c** Peripheral blood (PB) B cells were purified, as in Fig. 2A and stimulated (S) with CpG2006 +anti-Ig (A+G+M) Ab or vehicle (U) for 4 h. **a** *CCL3* and *TNF* mRNA expression was detected by qPCR. $*p < 0.05$. **b** CCL3 and TNF protein expression levels were assessed in the culture media by ELISA. $*p < 0.05$. **c** Conditioned medium (40% by volume) from B cell culture were co-cultured with human MSCs ± anti-CCL3 neutralizing Ab ± anti-TNF neutralizing Ab in OB-inducing medium for 3 days. ALP+ area was measured. $*p < 0.05$ vs. U, $\#p < 0.05$ vs. S-IgG. d PB B cells from RA patients and healthy controls (HC) were co-cultured with human MSCs in OB-inducing medium for 3 days. ALP staining was performed. $*p < 0.05$ vs. HC. **e** The expression levels of *ALP* and *Runx2* in hMSCs in **d** were measured by qPCR. Fold-changes were calculated by dividing patient values by the value from HC cells. Values represent individual HC/RA. $*p < 0.05$ vs. HC. **f** RA synovium was stained with Abs to CD20 (B cells), CCL3 and TNF. White arrows indicate the cells with dual staining. Bar = 25 μm. 3 or 6 patients and their controls were included in each experiment. All of these experiments were repeated at least once with representative images and quantifications shown. All error bars represent s.e.m. One way ANOVA followed by Dunnett's post-hoc multiple comparisons was performed for **c**. Two-tailed unpaired Student's *t*-test was performed for all the others

arthritis develops in the ankles of these mice when they are ~2-months-old and progresses with age leading to systemic bone loss and osteoporosis at ~4-months-old[47]. We used male TNF-Tg mice for all in vivo experiments because female TNF-Tg mice develop joint lesions earlier and die around 6-months-old due to lung inflammation[48]. (2) Collagen-induced arthritis (CIA). DBA/1 male mice (6–8-weeks-old) were purchased from Jackson Laboratories. CIA was induced according to a published protocol[49]. Briefly, chicken type II collagen (Chondrex, Redmond, WA, USA) was dissolved in 0.1M acetic acid overnight at 4 ℃. This was emulsified in an equal volume of complete Freund's adjuvant (Chondrex, Redmond, WA, USA). The mice were immunized intradermally at the base of the tail with 100 μL of emulsion containing 100 μg of type II collagen dissolved in phosphate-buffered saline (PBS) and boosted with the same dose on day 21. Mice were monitored and scored for the severity of arthritis every day, starting at 2 weeks after immunization. Arthritis typically begins between 3 and 6 weeks after immunization. We used CIA mice 2 weeks after the onset of arthritis. (3) TNF-Tg/CCL3 knockout (CCL3-KO),

TNF-Tg/TNF knockout (TNF-KO), TNF-Tg/CCL3-KO/TNF-KO, mTmG (mTmG mice possess loxP sites on either side of a membrane-targeted tdTomato cassette that is driven by the ROSA 26 promoter and express strong red fluorescence in all tissues and cell types) and GFP transgenic mice were used for ectopic bone formation experiments. TNF-Tg knockout animals were generated by crossing the TNF-Tg with CCL3-KO and TNF-KO mice, as detailed in the Supplementary Methods. All animals were randomized for their genotype information and were included in the study. All animal procedures were conducted in accordance with approved guidelines of the University of Rochester Committee for Animal Resources.

**μCT, histology, and histomorphometric analyses**. For μCT, knee joints were dissected free of soft tissue, fixed overnight in 10% buffered formalin, and scanned at high resolution (10.5 μm) on a VivaCT40 μCT scanner (Scanco Medical) using

300 ms integration time, 55 kVp energy, and 145 μA intensity. 3D images were generated using a constant threshold of 275 for all samples. For histology and histomorphometric analyses, knee and ankle joints were fixed in 10% buffered formalin, decalcified in 10% EDTA, embedded in paraffin, and sectioned at 4 μm thickness for 3 levels (50 μm apart). Sections were stained with H&E for routine histology and for TRAP activity to identify osteoclasts. The stained sections were digitized using a whole slide imaging system (Olympus VS120). Osteoclast numbers/mm bone surface, bone volume/tissue volume, inflamed area/tissue area and eroded surface/bone surface were analyzed on one section from each level using an osteomeasure image analysis system (Osteometrics)[8].

**Immunofluorescence staining**. (1) For mouse samples, knee or ankle joints were fixed in 10% buffered formalin, decalcified in 10% EDTA, and embedded in Tissue-Tek. Frozen sections (7 μm thick) were cut using a Leica CM1850 cryostat (Leica, German). For immunofluorescence (IF) staining, sections were blocked in PBS with 10% normal horse serum and 0.2% Triton X-100 for 1 h and then stained overnight with rabbit anti-osteocalcin (OCN) (ALX-210–333, Enzo, 1:200), rat anti-B220 (553086, BD Pharmingen, 1:50) or rabbit anti-CCL3 (AF-450-NA, R & D systems, 1:50) Abs at 4 ℃. After rinsing with PBS for 15 min, tissues were incubated with goat anti-rabbit Alexa Fluor 488 or goat anti-rat Alexa Fluor 568 (Invitrogen, 1:400) at room temperature. Slides were mounted with mounting medium containing DAPI (Vector), and images were taken with a fluorescence microscope (Axio IMAGER M1m, Zeiss). (2) Human RA synovial samples were fixed in 10% buffered formalin, embedded in paraffin, and sectioned (4 μm thick). Sections were incubated at 60℃ for one hour, transferred to xylene and hydrated by sequential immersion in ethanol, 95% ethanol, 75% ethanol and finally water. Antigens were unmasked in DAKO antigen retrieval solution (S1699, DAKO) for 30 min. Non-specific binding was blocked with 5% normal donkey serum (017-000-121, Jackson ImmunoResearch Laboratories) for 30 min at room temperature. Mouse anti-human CD20 (GTX29475, Gene Tex, 1:100), rabbit anti-human CCL3 (PA1-38160, Thermo Fisher Scientific, 1:100), or rabbit anti-human TNF (ab9739, Abcam, 1:100) Abs were added to the slides and incubated at room temperature in a humid chamber overnight. On the next day, slides were washed with PBS, and Biotin-donkey anti-mouse IgG (715-066-150, Jackson ImmunoResearch Laboratories, 1:200), Streptavidin-Cyanine 5 (19-4317-82, eBioscience, 1:200), or FITC-donkey anti-rabbit IgG (711-066-150, Jackson ImmunoResearch Laboratories, 1:200), were added and incubated for 2 h at room temperature. Slides were washed with PBS and mounted with Prolong gold antifade with DAPI (P36931, Life Technologies). Representative pictures were taken with a Zeiss Axiocam digital Camera (Carl Zeiss) and a Zeiss Axioplan 2 microscope.

**B cell isolation**. B cells were isolated from bone marrow (BM), subchondral BM (SBM) and synovium of mice. BM cells were flushed out from the femora and tibiae. To harvest SBM cells, all the joints including knees, fore and hind paws were separated, and muscles were removed. Subchondral (between the articular cartilage and growth plate) and synovial tissues were cut into small pieces and digested using Accumax Cell Dissociation Solution (Innovative Cell Technologies) at room temperature for one hour to yield SBM cells. Both SBM and synovial B cells were isolated together given their anatomic proximity and are referred to throughout the manuscript as SBM (SBM plus synovial B cells). RBCs from BM and SBM cells were eliminated using RBC lysis buffer. B cells were then purified, based on CD19 expression using microbeads conjugated with anti-CD19 antibody (Ab) and magnetic isolation (Miltenyi Biotec, Auburn, CA)[50]. Human peripheral blood mononuclear cells (PBMCs) were purified from heparinized PB by Ficoll-Hypaque density gradient centrifugation (Pharmacia Biotech), and B cells were isolated based on CD19 expression using CD19 microbeads and magnetic isolation (Miltenyi Biotec, Auburn, CA)[4].

**Cell cultures**. (1) For B cell/OB co-cultures, purified BM or SBM B cells and mouse bone-derived mesenchymal progenitor cells (MPCs) were used. Briefly, long bones were flushed several times with PBS, cut into small pieces, and cultured in a plastic dish for 3 days. The bone pieces were transferred into a clear dish as passage 1 and continually cultured for another 7 days to allow cell growth to confluence. Third passage MPCs were used[51]. Human mesenchymal stem cells (hMSCs) were purchased from Lonza (Allendale, NJ, catalogue number PT-2501). MPCs or hMSCs were plated in 96-well-plates at $3 \times 10^3$ per well for 24 h. Mouse or human B cells ($3 \times 10^4$) were then seeded on to MPCs or hMSCs and co-cultured in OB-inducing medium containing 50 μg/ml ascorbic acid and 10 mM β-glycerophosphate for 2 or 3 days. ALP staining was performed using the 1-step NBT/BCIP reagent (Thermo Scientific). (2) For murine B cell stimulation, purified B cells were cultured with 2.5 μg/ml anti-CD40 (553722, BD Biosciences) plus 10 ng/ml IL4 (404-ML-010, R & D systems) and 10 μg/ml LPS (L4391-1MG, Sigma) or PBS control for 24 h. At the end of the culture period, conditioned medium was collected for co-culture or ELISA. (3) For CFU-F and CFU-ALP colony formation assays, BM cells were cultured in 12-well-plates at $2 \times 10^6$ cells per well in α-MEM containing 10% FCS (Hyclone Laboratories) with or without 50 μg/ml ascorbic acid and 10 mM β-glycerophosphate for 12 days. At the end of the culture period, cells were stained for CFU-F or CFU-ALP activity and positive colonies (>20 cells in a single colony) were counted. (4) For bone nodule formation, BM cells were cultured in α-MEM

containing 10% FCS for 7 days to generate BM-MPCs. BM-MPCs were then cultured in OB-inducing medium for another 14 days and mineralized bone nodules were counted after von Kossa staining. (5) For osteoclastogenic assays, BM cells were cultured with conditioned medium (1:50 dilution) from a M-CSF-producing cell line for 3 days in α-MEM with 10% FCS to enrich for osteoclast precursors (OCPs), which were then cultured with M-CSF-conditioned medium and RANKL (10 ng/ml, R&D) for 2–3 days. After multinucleated cells were observed under a microscope, the cells were fixed, stained for TRAP activity to identify OCs (TRAP+ cells containing >3 nuclei) and counted[52]. (6) For blocking experiments with neutralizing Abs, anti-mouse CCL3 Ab (AF-450-NA), anti-mouse TNF Ab (AF-410-NA), anti-human CCL3 Ab (MAB270), and anti-human TNF Ab (MAB610), were purchased from R & D systems and used according to the manufacturer's instructions. (7) For human B cell stimulation, purified B cells were cultured with 5 μg/ml CpG 2006 (Oligos Etc.) plus 10 μg/ml Anti-Ig (A+G +M) (109-006-064, Jackson ImmunoResearch Laboratories) or PBS control for 4 h. At the end of the culture period, conditioned medium was collected for co-culture or ELISA.

**RNA-Sequencing and data analysis**. B cells were purified from BM of 6-m-old WT (WT BM, n=3) and TNF-Tg (TNF-Tg BM, $n = 4$) mice and from subchondral BM of TNF-Tg (TNF-Tg SBM, $n = 4$) with anti-CD19 beads. The purity of B cells was confirmed by flow cytometric analysis using anti-B220 and anti-CD19 Abs. Following purification, RNA was harvested and subjected to low input RNA-seq (purity and workflow in Supplementary Fig. 12) (NCBI Sequence Read Archive; accession no. SRP157127). Briefly, mRNA was extracted from purified B cells ($1 \times 10^4$) using a RNeasy Plus Micro Kit (QIAGEN) and subjected to RNA-sequencing (seq) using a low input protocol (Clontech SMARTer Technology). Sequenced reads were cleaned according to a pre-processing workflow (Trimmomatic-0.32) before mapping them to the Mus musculus genome (mm10) with SHRiMP2.2.3 (http://compbio.cs.toronto.edu/shrimp). Subsequently, cufflinks 2.0.2 (cuffdiff2) was used to perform differential gene expression analysis with an FDR cutoff of 0.05 (95% confidence interval). Data visualization of RNA-seq results was accomplished using Matlab (The Mathworks Inc., Natick MA).

**Quantitative real-time RT-PCR**. Total RNA was extracted using TRIzol reagent (Invitrogen). cDNAs were synthesized using a iSCRIPT cDNA Synthesis Kit (Bio-Rad). Quantitative real-time RT-PCR amplifications were performed in an iCycler (Bio-Rad) real-time PCR machine using iQ SYBR Green supermix (Bio-Rad), according to the manufacturer's instructions. *Gapdh* was amplified on the same plates and used to normalize the data. Each sample was prepared in triplicate and each experiment was repeated at least three times. The relative abundance of each gene was calculated by subtracting the CT value of each sample for an individual gene from the corresponding CT value of *Gapdh* (ΔCT). ΔΔCT were obtained by subtracting the ΔCT from the reference point. These values were then raised to the power 2 ($2^{\Delta\Delta CT}$) to yield fold-expression relative to the reference point. Representative data are presented as means + SD of the triplicates or of four wells of cell cultures. The sequences of primer sets mCCL3, mTNF, mDkk3, mRunx2, mALP, mGapdh for mouse and hCCL3, hTNF, hRunx2, hALP, hGapdh for human mRNA are shown in Supplementary Table 3.

**Western blot analysis and ELISA**. Whole-cell lysates (30 μg protein/lane) were loaded in 10% SDS-PAGE gels and immunoblotted with Abs to p-AKT, Total-AKT, p-ERK, Total-ERK, p-IκBα, p52, Total β-catenin, Active β-catenin (Cell Signaling Technology), and RelA, RelB, p50, β-actin (Santa Cruz Biotechnology Inc.). Conditioned medium was collected from B cell culture and subjected to ELISA analysis. Mouse TNF ELISA kit (88-7324-22), mouse CCL3 ELISA kit (88-56013-22), human TNF ELISA kit (88-7346-22), and human CCL3 ELISA kit (88-7035-22), were purchased from Invitrogen and used according to the manufacturer's instructions.

**In vivo ectopic bone formation assay**. MPCs were isolated from WT or GFP transgenic mice and cultured to a third passage. BM B cells from WT, TNF-Tg, TNF-Tg/CCL3-KO, TNF-Tg/TNF-KO, TNF-Tg/CCL3-KO/TNF-KO or mTmG mice were purified with anti-CD19 beads. Gelfoam (Pfizer) was loaded with $1 \times 10^6$ MPCs and $1 \times 10^7$ B cells and implanted into the dorsal subcutaneous tissue of 2-month-old NOD-SCID mice[31]. At 4 weeks after implantation, the implants were harvested, processed through paraffin and sections were stained with H&E and Goldner's Trichrome.

**B cell depletion in mice**. 2.5-month-old TNF-Tg mice were given 10 mg/kg murine anti-CD20 (18B12amCD20 from Biogen) Ab by retro-orbital injection weekly for eight weeks ($n = 5$). A control group was given 10 mg/kg isotype control (ZB8msIgG2a) ($n = 5$). After eight weeks, right knees and ankles were subjected to μCT and histologic analysis. BM cells were collected from left legs and combined together in each group for CFU-F, CFU-ALP, bone nodule, and osteoclast cultures. Depletion of B cells in spleen was assessed by flow cytometry[32], and in bone sections by IF.

**Flow cytometry and cell sorting**. Cells were harvested and red blood cells were lysed. Cells were stained with various Abs for 30 min and subjected to 12-color LSRII (BD Biosciences, San Jose, CA) for flow cytometric analysis, or FACS Aria II (BD Biosciences, San Jose, CA) for cell sorting. Results were analyzed by Flowjo7 data analysis software (FLOWJO, LLC Ashland, OR).

**Patients and sampling**. Detailed written informed consent was obtained from all patients and healthy donors in accordance with protocols approved by the Human Subjects Institutional Review Board of the University of Rochester Medical Center. PBMCs were obtained from CCP+ RA patients fulfilling 1987 American College of Rheumatology diagnostic criteria (patients on prior or current biologic use were excluded) and from age- and gender-matched healthy controls under informed consent. RA patients (n=6) had a mean age of 56 years, mean disease duration of 8.2 years, and evidence of disease activity, based on the presence of swollen and tender joints. Disease activity was accessed by DAS28 scores (mean, 3.5; range, 2.24 to 5.31). PBMCs were isolated from heparinized PB by Ficoll-Hypaque density gradient centrifugation (Pharmacia Biotech)[4]. Synovial samples were obtained from 3 RA patients undergoing elective wrist joint surgery for a variety of indications, most commonly carpal tunnel release[53]. All patients were CCP+ with a range of disease duration from 2 to 8 years. Synovitis scores ranged from 8 to 13 of a possible 20, indicating moderate to high RA-associated synovitis (histological grading based on the presence and degree 0–4 of synovial lining hyperplasia, cellularity of the synovial stroma, extent of inflammatory infiltrate, number of lymphoid follicles, and the degree of vascularity)[54].

**Statistical analysis**. All in vitro experiments were performed at least 3 times and in vivo experiments were performed twice. Statistical analysis was performed using GraphPad Prism 5 software (GraphPad Software Inc., San Diego, CA, USA). Comparisons between 2 groups were analyzed using the 2-tailed unpaired Student's $t$ test. Comparisons among 3 or more groups were carried out using one way ANOVA followed by Dunnett's post-hoc multiple comparisons. $p$ values <0.05 were considered statistically significant.

## Data availability

The raw sequencing data that support the findings of this study have been deposited in 'NCBI Sequence Read Archive' under the accession no. 'SRP157127'. The authors declare that all other data supporting the findings of this study are available within the article and its Supplementary Information files are available from the corresponding author upon request.

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

## Acknowledgements

We thank Drs. Robert Dunn and Marilyn Kehry from Biogen for providing murine anti-CD20 and isotype control Abs, Mr. Martin Chang for technical assistance with the whole slide-scanner and Dr. Mengmeng Wang for technical assistance with the cell culture. We thank the Functional Genomics and Flow Cytometry Cores for their technical support. Research was supported by grants from National Institute of Health PHS awards (AR63650, and AR69789 to L.X.; AR43510 and AG049994 to B.F.B.; AI563262, AI078907, and AR071670 to J.H.A.). L.X. is also supported by an award from NYSTEM N13G-084 (C029548). J.H.A. is also supported by the NIAMS Accelerating Medicines Partnership (1UH2AR067690), and the Bertha and Louis Weinstein research fund. W.S. is supported by the grants from the National Natural Science Foundation of China (81670965) and from Natural Science Foundation of Jiangsu Province in China (BK20180034). Some experiments were performed by the CMSR cores (microCT) or using CMSR core equipment (frozen sectioning and whole slide imaging), which are supported by grants from National Institute of Health PHS awards (1S10RR027340-01 to B.F.B., P30 AR069655 to E.M.S.).

## Author contributions

Study design: W.S., N.M., H.Z., B.F.B., J.H.A., and L.X. Study conduct: W.S., N.M., A.R., J.R.M., V.W., J.G., T.O., X.Z., H.Z. Data collection: W.S., N.M. Data analysis: W.S., N.M., A.R. Data interpretation: W.S., N.M., H.Z., B.F.B., J.H.A. and L.X. Drafting manuscript: W.S., N.M., J.H.A., and L.X. Revising manuscript content: W.S., N.M., A.R., B.F.B., J.H.A., and L.X. Approving final version of manuscript: W.S., N.M., A.R., J.R.M., V.W., J.G., T.O., X.Z., H.Z., B.F.B., J.H.A., and L.X. J.H.A. and L.X. take responsibility for the integrity of the data analysis.

## Additional information

**Competing interests:** The authors declare no competing interests.

