## [Peer Review File · Nature Communications]

Reviewers' comments:

Reviewer #1 (Remarks to the Author):

Sun and colleagues investigated the role of B cells on osteoblasts. They show that bone marrow B cells from arthritic mice suppress osteoclast differentiation through a CCL3- and TNF-dependent mechanism. They also found that depletion of B cells by anti-CD20 antibody in arthritic mice increases osteoblasts counts while decreasing osteoclasts. Overall this is an interesting paper from a group with long-standing expertise in arthritis Research.

Some points need to be considered:

The rather huge difference in TNF and CCL3 expression between B cells localized in the subchondral bone marrow and those localized in the non-subchondral bone marrow is surprising. This observation raises the question of the nature of these B cells in the subchondral bone marrow. Are these cells plasmablasts or plasma cells, which have been re-circulating from the secondary lymphatic organs into the bone marrow? It appears that they do not resemble the „normal“ bone marrow B cells.

As the TNFtg model is not necessarily an autoimmune model associated with autoantibody formation one questions whether there is a change in the peripheral, splenic and bone marrow B cell composition during this form of arthritis.

The observation that TNF is one of the factors produced by B cells which inhibits osteoblast formation is interesting. However, in vivo, the transgenic source of TNF may not only be the B cells and hence suppression of osteoblasts in the TNFtg model is likely to be based on the cumulative effect of all cells bearing the TNF transgene.

CD20-targeted B cell depletion led to an increase in osteoclasts and a decrease in osteoclasts. However, in Figure 5 it is not shown whether this treatment also affected arthritis.

So far, data on the effects of B cells on bone have not been supporting an osteoblast phenotype with the exception of an intrinsic osteopenic phenotype of Pax5 mutants while no bone phenotype was observed in B cell deficient mMT and Rag1 mice. Although this observation may be explained by the simultaneous absence of B cells negatively (such as shown here) and positively (such most likely B10 cells) impacting bone, this point should be discussed in more detail.

Minor:

Figure 5C: It is hard to distinguish bone effects between IgG and aCD20 in the knee. While the differences in the patella are convincing, this is a somewhat unusual localization for quantifying bone damage in arthritis.

Typo page 7 line 2

Reviewer #2 (Remarks to the Author):

In this manuscript, Wen Sun and colleagues explore the interaction between B cells and osteoblasts in the context of arthritis. Their starting point is the observation that B220+ cells localize near osteoblasts in subchondral bone marrow of femora and tibiae, as well as in synovium, in TNF-Tg mice that have already developed severe arthritis and systemic bone loss. Following this information, they performed transcriptomic analyses comparing B cells from the subchondral bone marrow and bone marrow of WT and TNF-Tg mice. This led them to observe increased expression of CCL3, TNF, and Dkk3, which can act as osteoblast inhibitors, in B cells from the subchondral bone of TNF-Tg mice. Based on this, the authors analysed the capacity of B cells to influence the development of osteoblasts in vitro. B cell conditioned-medium reduced the development of osteoblasts, and this could be partially controlled upon neutralization of CCL3 or TNF. To support the proposition that this pathway is relevant in vivo, the authors provide data showing that B cell-depletion therapy leads to increases in numbers of osteoblasts in TNF-Tg mice. Finally, human peripheral blood B cells activated in a way inducing their expression of CCL3 and TNF displayed an inhibitory effect on osteoblast development. In total, the manuscript provides an interesting hypothesis on the interaction between B cells and bone-generating cells in arthritis. However, it lacks the final demonstration that this interaction actually takes place in vivo. The presentation of some data is suboptimal, and with very small group sizes. The English is in several places unclear. For instance, the one sentence summary is unclear. The first sentence of the abstract is also incorrect. These aspects should be addressed before the manuscript can be considered for publication in Nature Communications. I provide below a detailed list of comments:

Figure 1: The quality of the immunofluorescence data shown in panel A is very poor. It is important to stress that B cells and osteocalcin-expressing cells do not appear to be in direct contact in the images provided. It is therefore unclear whether these cells can directly communicate in vivo. The terminology adjacent is ambiguous. There is no indication of how the quantifications on tissue sections are done. What do n=5 reflect? Are these five mice or five measurements in different bone areas? The authors need to clarify this. How many independent experiments were performed? How many mice were included in each experiment, and how were the quantifications performed? Considering that an infiltration of B cells in bone was already documented in the context of arthritis, the authors should further document the phenotype(s) of the B cells that accumulate in these areas. Are these antibody-secreting cells? Do they proliferate locally?

Figure 2: The comparison through transcriptomic analyses of B cells from different bone regions is interesting. However, the purity of the B cells isolated from the SBM is not clear. What is the purity of the B cell fractions used in these analyses? How was the sequencing done? The experimental layout for these transcriptome analyses is also not clear. Was the transcriptome analysis done on single sample or in replicates? If so, how many samples were independently analysed? Were mice pooled for these analyses? How was the identification of differentially expressed genes done? The authors should deposit the raw data of these transcriptome analyses in a publicly available database. They should also provide tables with at least the 20 most differentially expressed genes for each comparison. Do the authors also find expression in B cells of other molecules relevant for bone homeostasis such as BMP-7 in these transcriptome analyses? B cells were previously identified as the predominant cell type expressing BMP-7 in BM infiltrate (Görtz et al. Journal of Bone and Mineral Research 2004). A global analysis of the transcriptome results should also be presented to illustrate the pathways most significantly relevant for the differences between these various B cell fractions. What does n=4 mean for the qPCR data? Does it mean that 4 mice were analysed individually? If so, this is a very limited sample size. Does it come from a single experiment? When were B cells isolated from the mice for analysis? Similarly, when were B cells isolated from mice with CIA? What does n=6 mean in this case? Does it mean that 6 mice were analysed individually? Since the authors can isolate B cells, it would be highly relevant that they characterize these cells by flow cytometry in order to document which B cell subsets are involved in the process described by the authors.

Figure 3: An important control with conditioned medium containing LPS, aCD40, and IL4 but no B cells, are missing in panel A. The fact that untreated B cells have an effect would suggest that TNF is involved in the observed phenomenon. How much TNF is present in the B cell culture supernatant? What do the 1:1, 1:2, 1:4, and 1:8 conditions indicate? Can the authors confirm that the B cells were stimulated for 24h, and that the CM was collected at that time point. This should be indicated in the figure legend.

Panel B: Were the B cells used in these co-cultures activated prior to the co-culture?

The authors mention in the result section that AKT and ERK proteins belong to the same pathway. This is incorrect. These are two different pathways that cross-talk with each other.

Panel F: which lane correspond to the co-culture with WT versus TNF-Tg B cells? How were cells obtained from the cultures? Were B cells removed from these co-cultures to perform the western blot specifically using MPC?

Figure 4: It is not clear whether the B cells were cultivated or not prior to the co-culture with MPC.

Panel A: What does n=4 mean? Do the data show a compilation of 4 independent experiments or a representative experiment? This should be better explained also for the other figure panels. The authors show that adding anti-CCL3 or anti-TNF antibodies to the culture increases the ALP area. The effects are however incomplete in both cases. Would neutralization of both factors lead to a complete abrogation of the effect of the B cells? I could not find the references of the reagents used to neutralize CCL3 or TNF in the Materials and Methods. This should be completed. The data shown in Supplementary Fig 3 on the effect of CCL3 on MPC cultures, and on the expression of CCL3 by B cells in situ are interesting. It might be relevant to move some of these data to main figures. It would be useful that the authors perform a different staining to confirm that B220+ cells are indeed B cells. B220 is not a strict B cell-specific marker. Staining in addition for immunoglobulin (eg Igk) would provide further strength to these data.

Figure 5: panel B: the authors should explain how they made the quantification of the osteoblasts. The distribution of the cells is not uniform. Do the data actually show number of OB per mm, or per mm²?

Taken together, the data shown in Fig 4 and 5 suggest that B cells control OB formation through production of CCL3 and TNF. However, the effect of the 8-weeks B cell-depletion treatment could affect OB via many different ways others than this one. Thus, the data do not demonstrate directly that B cell production of CCL3 or TNF affects OB in vivo. It is critical that the authors demonstrate this using a direct approach, especially since direct contact between OB and B cells was not obvious in histology. This should be done using mixed BM chimera in which only B cells cannot produce CCL3 or TNF.

Figure 6: The reagents used to activate B cells should be better described, with indication of company name and catalog number. The name of the clones should be provided for the antibodies. Were B cells from HD and RA patients used in panel D activated prior to their use?

In supplementary figure 6, the authors claim that memory B cells are the most relevant B cell subset based on their CD27 expression. However, antibody secreting cells also express high levels of CD27. How can the authors distinguish between the involvement of memory versus antibody-secreting cells?

Point to point responses

Reviewers' comments:

Reviewer #1 (Remarks to the Author):

Sun and colleagues investigated the role of B cells on osteoblasts. They show that bone marrow B cells from arthritic mice suppress osteoblast differentiation through a CCL3- and TNF-dependent mechanism. They also found that depletion of B cells by anti-CD20 antibody in arthritic mice increases osteoblasts counts while decreasing osteoclasts. Overall this is an interesting paper from a group with long-standing expertise in arthritis Research.

Some points need to be considered:

- 1) The rather huge difference in TNF and CCL3 expression between B cells localized in the subchondral bone marrow and those localized in the non-subchondral bone marrow is surprising. This observation raises the question of the nature of these B cells in the subchondral bone marrow. Are these cells plasmablasts or plasma cells, which have

been re-circulating from the secondary lymphatic organs into the bone marrow? It appears that they do not resemble the "normal" bone marrow B cells.

- This is an important question. We have now included detailed flow cytometry analysis of bone marrow and subchondral bone marrow cells from TNF-Tg mice for comparison (and WT-BM) in supplemental figure 2. We discuss these findings in the revised manuscript to address the reviewer's concern on p.8.

Direct comparison by flow cytometry of B cells in the TNF-Tg mouse in the spleen, blood, total bone marrow, and synovium/subchondral bone marrow. The synovium/subchondral BM B cell distribution is distinct compared to total BM with an enrichment of more mature B cells (80% AA4.1- compared to 40% of the total BM B cells in the red rectangle, 50% vs. 30% IgD+, 70% vs. 35% IgM+).

We also have taken the liberty of reproducing some of this data below with inclusion of splenic B cells so that the reviewer can more directly compare the B cell phenotypes in the different compartments in the TNF-Tg. Although the subchondral bone marrow/synovial B cells are diverse in phenotype, the majority are mature B cells (IgM+IgD+AA4.1-). Of note, the phenotype of the subchondral B cells is quite distinct from total BM B cells, the latter enriched for AA4.1+IgD- precursor cells. We have now also repeated an experiment with a new cohort of TNF-Tg mice to more directly examine expression of the plasma cell marker CD138 in the subchondral B cells and demonstrate that only a very small frequency of the B cells are plasma cells (<5-10%) (supplemental figure 2Ciii; example dot plot below). Additionally, the transcriptome analysis of the CD19+ B cells isolated from the subchondral bone marrow/synovium does not demonstrate a typical plasma cell signature, such as enrichment for particular genes like XBP1 or BLIMP1 or pathways such as the unfolded protein response and intracellular protein transport (Shi W, Liao Y, Willis SN, Taubenheim N, Inouye M, Tarlinton DM, et al. Transcriptional profiling of mouse B cell terminal differentiation defines a signature for antibody-secreting plasma cells. Nat Immunol (2015) 16: 663–673). It remains an important question whether the ‘pathogenic’ B cells in the subchondral BM are generated elsewhere and recirculate to the joint. Based on the transcriptome analysis, we speculate that the joint microenvironment plays a critical role in promoting a pathogenic B cell function. The finding of local B cell proliferation (see Review 2) further supports this contention. These important considerations are now expanded in the revised discussion.

Flow cytometry analysis of SBM cells from TNF-Tg mice reveals that most of the B cells are not plasma cells. To harvest SBM cells, all the joints including knees, fore and hind paws were separated, and muscles were removed. Subchondral (between the articular cartilage and growth plate) and synovial tissues were cut into small pieces and digested using Accumax Cell Dissociation Solution (Innovative Cell Technologies) at room temperature for one hour to yield SBM cells. RBCs from BM and SBM cells were eliminated using RBC lysis buffer. B220+CD19+ B cells were gated and the expression of CD138 and MHCII examined to define plasma cells (CD138+MHCII low) and plasmablasts (CD138+MHCII high). The dot plot represents the overlay of 4 separate mice, with the % +/- SEM.

2) *As the TNFtg model is not necessarily an autoimmune model associated with autoantibody formation one questions whether there is a change in the peripheral, splenic and bone marrow B cell composition during this form of arthritis.*

-Although TNF-Tg disease is not autoantibody driven, it displays many of the histopathological findings of human RA (synovial hyperplasia, inflammatory infiltrates, and joint

erosion), other signs of systemic inflammation, and is clearly B cell dependent. However, we did not see significant differences in the composition of B cell subsets in the spleen and bone marrow compared to WT mice (supplemental figure 2). On the other hand, the function of the B cells in both the total bone marrow and subchondral BM is clearly abnormal in the TNF-Tg, as evidenced by the higher expression of OB inhibitory factors (Tg-SBM>Tg-BM>WT-BM). Additionally, these features were recapitulated in a 2nd murine mouse model (CIA) and human RA. Thus, B cells in the target tissue (synovium and SBM) are skewed toward a pathogenic phenotype. The role of autoantibodies in the disease process is certainly an important question which we have not directly addressed. On the other hand, our data supports the important role of B cells in the disease process by antibody-independent mechanisms (cytokine and chemokine production).

3) *The observation that TNF is one of the factors produced by B cells which inhibits osteoblast formation is interesting. However, in vivo, the transgenic source of TNF may not only be the B cells and hence suppression of osteoblasts in the TNFtg model is likely to be based on the cumulative effect of all cells bearing the TNF transgene.*

- This is an important point. Indeed, the transgenic source of TNF is by no means limited to the B cells. To address this concern, we have now performed ectopic bone formation experiments using WT MPCs and BM B cells from WT, TNF-Tg, TNF-Tg/CCL3-KO, TNF-Tg/TNF-KO or TNF-Tg/CCL3-KO/TNF-KO mice, as shown in revised Fig. 5F-H. The data revealed that when transplanted with WT MPCs, TNF-Tg BM B cells mediated a decrease in bone volume compared with WT BM B cells, while B cells from TNF-Tg/CCL3-KO BM or TNF-Tg/ TNF-KO BM were partly rescued from this OB inhibitory phenotype (Fig. 5F-H). Furthermore, TNF-Tg/CCL3-KO/TNF-KO BM B cells plus WT MPCs had higher bone volume compared with B cells from TNF-Tg/CCL3-KO BM or TNF-Tg/ TNF-KO BM, and had no difference compared to WT BM B cells (Fig. 5F-H). These data suggest that the OB inhibitory effect of RA is critically mediated by B cell production of TNF and CCL3 and extend our prior findings to an in vivo model.

4) *CD20-targeted B cell depletion led to an increase in osteoblasts and a decrease in osteoclasts. However, in Figure 5 it is not shown whether this treatment also affected arthritis.*

-We now include new analyses in supplemental figure 8 demonstrating that BCDT decreased the area of inflammation and bone erosion in the knees and ankles of TNF-Tg mice

in addition to the observed effects on osteoclasts and osteoblasts. It is possible that the OC and OB changes are not all directly mediated by the absence of B cells but rather indirectly by inflammation reduction in the absence of B cells. This makes the ectopic bone formation assay discussed above an important complementary *in vivo* experiment.

5) *So far, data on the effects of B cells on bone have not been supporting an osteoblast phenotype with the exception of an intrinsic osteopenic phenotype of Pax5 mutants while no bone phenotype was observed in B cell deficient uMT and Rag1 mice. Although this observation may be explained by the simultaneous absence of B cells negatively (such as shown here) and positively (such most likely B10 cells) impacting bone, this point should be discussed in more detail.*

- This is an excellent point and we have expanded our discussion of this topic on p. 17.

Minor:

Figure 5C: It is hard to distinguish bone effects between IgG and aCD20 in the knee. While the differences in the patella are convincing, this is a somewhat unusual localization for quantifying bone damage in arthritis.

- We have replaced the picture of the tibial SB in Figure 5C (now Figure 6C) with another representative image. Of note, micro-CT analysis reproducibly demonstrates that tibial SB bone volume increased in the B cell depletion group compared with the control IgG group. Others have demonstrated larger differences in patella volumes after anti-CD20 treatment (*J Immunol.* 2010 Jun 1;184(11):6142-50) which may be related to greater B cell accumulation in the patella (Fig. S1A).

Typo page 7 line 2

- We corrected this mistake.

Reviewer #2 (Remarks to the Author):

In this manuscript, Wen Sun and colleagues explore the interaction between B cells and osteoblasts in the context of arthritis. Their starting point is the observation that B220+ cells localize near osteoblasts in subchondral bone marrow of femora and tibiae, as well as in synovium, in TNF-Tg mice that have already developed severe arthritis and systemic bone loss. Following this information, they performed transcriptomic analyses comparing B cells from the subchondral bone marrow and bone marrow of WT and TNF-Tg mice. This led them to observe

increased expression of CCL3, TNF, and Dkk3, which can act as osteoblast inhibitors, in B cells from the subchondral bone of TNF-Tg mice. Based on this, the authors analysed the capacity of B cells to influence the development of osteoblasts *in vitro*. B cell conditioned-medium reduced the development of osteoblasts, and this could be partially controlled upon neutralization of CCL3 or TNF. To support the proposition that this pathway is relevant *in vivo*, the authors provide data showing that B cell-depletion therapy leads to increases in numbers of osteoblasts in TNF-Tg mice. Finally, human peripheral blood B cells activated in a way inducing their expression of CCL3 and TNF displayed an inhibitory effect on osteoblast development. In total, the manuscript provides an interesting hypothesis on the interaction between B cells and bone-generating cells in arthritis. However, it lacks the final demonstration that this interaction actually takes place *in vivo*. The presentation of some data is suboptimal, and with very small group sizes. The English is in several places unclear. For instance, the one sentence summary is unclear. The first sentence of the abstract is also incorrect. These aspects should be addressed before the manuscript can be considered for publication in Nature Communications. I provide below a detailed list of comments:

- We appreciate the careful critiques of the reviewer. The manuscript has now been very carefully edited to correct language discrepancies. We have also now highlighted in the figure legends where experiments were repeated multiple times *in vivo* to confirm findings. We acknowledge the concern that the original submission lacked the 'final' *in vivo* demonstration of a functionally relevant B cell-OB interaction. Addressing the latter concern required generation of TNF-Tg mice on a CCL3 knock-out background and TNF knock-out background (and double knock-outs), followed by transplantation of B cells from the 3 strains (or the WT TNF-Tg control) into SCID mice with normal MPCs in order to define the effects of specific deletion of TNF and/or CCL3 in the B cell compartment.

1) Figure 1: The quality of the immunofluorescence data shown in panel A is very poor. It is important to stress that B cells and osteocalcin-expressing cells do not appear to be in direct contact in the images provided. It is therefore unclear whether these cells can directly communicate *in vivo*. The terminology adjacent is ambiguous. There is no indication of how the quantifications on tissue sections are done. What do n=5 reflect? Are these five mice or five measurements in different bone areas? The authors need to clarify this. How many independent experiments were performed? How many mice were included in each experiment, and how were the quantifications performed? Considering that an infiltration of B cells in bone was already documented in the context of arthritis, the authors should further

document the phenotype(s) of the B cells that accumulate in these areas. Are these antibody-secreting cells? Do they proliferate locally?

Frozen sections of knees from 6-m-old TNF-Tg were stained with anti-B220 Ab for B cells (red) and anti-osteocalcin (OCN) Ab for OBs (green). White arrows = B cells adjacent to OCN+ cells.

-We regret the lack of clarity in our original submission. We have now included higher resolution images in Figure 1A and a higher magnification of SBM areas from TNF-Tg mice (Figure 1B) (reproduced above in a larger field) with white arrows highlighting the B cell-OB interactions. Another important point is that the number of OBs is clearly reduced in the setting of increased B cell infiltrates (Figure 1C). We have also clarified the method of morphometric analysis in the figure legends and the fact that n=5 signifies 5 mice in the data shown from one independent experiment. However, every experiment was repeated at least once. We have addressed the question of B cell phenotype in Review 1. We appreciate the reviewer's suggestion to further analyze the B cells in situ for evidence of local proliferation. Indeed, the SBM B cells in the TNF-Tg mouse are Ki67+. In summary, based on flow cytometry, IHC, and transcriptome analysis the B cells infiltrating the synovium and subchondral BM are heterogeneous but dominated by mature B cells (as opposed to plasma cells) that appear to be

locally activated and proliferating.

B cells in the subchondral BM of TNF-Tg mice have increased proliferation.

Frozen sections of leg including knee joints from 5-m-old TNF-Tg and WT mice were subjected to IF with anti-B220 Ab for B cells (red) and anti-Ki67 Ab for cell proliferation (green). There are multiple areas of dual staining as highlighted by the arrow in one section. 5 mice and their controls were included in each experiment. Representative data is shown from TNF-Tg and control mice.

- 2) *Figure 2: The comparison through transcriptomic analyses of B cells from different bone regions is interesting. However, the purity of the B cells isolated from the SBM is not clear. What is the purity of the B cell fractions used in these analyses? How was the sequencing done? The experimental layout for these transcriptome analyses is also not clear. Was the*

transcriptome analysis done on single sample or in replicates? If so, how many samples were independently analysed? Were mice pooled for these analyses? How was the identification of differentially expressed genes done? The authors should deposit the raw data of these transcriptome analyses in a publicly available database. They should also provide tables with at least the 20 most differentially expressed genes for each comparison. Do the authors also find expression in B cells of other molecules relevant for bone homeostasis such as BMP-7 in these transcriptome analyses? B cells were previously identified as the predominant cell type expressing BMP-7 in BM infiltrate (Görtz et al. Journal of Bone and Mineral Research 2004). A global analysis of the transcriptome results should also be presented to illustrate the pathways most significantly relevant for the differences between these various B cell fractions. What does n=4 mean for the qPCR data? Does it mean that 4 mice were analysed individually? If so, this is a very limited sample size. Does it come from a single experiment? When were B cells isolated from the mice for analysis? Similarly, when were B cells isolated from mice with CIA? What does n=6 mean in this case? Does it mean that 6 mice were analysed individually? Since the authors can isolate B cells, it would be highly relevant that they characterize these cells by flow cytometry in order to document which B cell subsets are involved in the process described by the authors.

- The methodology for RNA-sequencing is now provided in more detail in the Methods. We have added flow analysis documenting the purity of the B cells and the workflow for the RNA-seq in Fig S10. We have deposited the raw data of RNA-Seq at the NCBI Sequence Read Archive under the accession no. SRP157127. A table with the 30 most differentially expressed genes is now provided in Fig S3 & S4, as well as pathway analysis (Fig S5). The question of whether BMP-7 is differentially expressed in the TNF-Tg joint infiltrating B cells is an interesting one. Although BMP-7 was higher in TNF-Tg SBM B cells vs. TNF-Tg BM B cells, this did not reach statistical difference ($p=0.13$). N=4 does indicate the number of mice examined with matched controls for the RNA sequencing experiment in 6-month-old TNF-Tg mice. The PCR validation in Fig 2C and 2D was performed on an independent cohort of n=4-6 mice per group and repeated at least once (representative experiment shown). CIA mice were studied 6 weeks after the onset of arthritis. We have now included this information in both the Figure legends and Methods. B cells were characterized by flow cytometry and that data is now included in Fig S2.

3) *Figure 3: An important control with conditioned medium containing LPS, aCD40, and IL4 but no B cells, are missing in panel A. The fact that untreated B cells have an effect would*

suggest that TNF is involved in the observed phenomenon. How much TNF is present in the B cell culture supernatant? What do the 1:1, 1:2, 1:4, and 1:8 conditions indicate? Can the authors confirm that the B cells were stimulated for 24h, and that the CM was collected at that time point. This should be indicated in the figure legend.

Panel B: Were the B cells used in these co-cultures activated prior to the co-culture?

The authors mention in the result section that AKT and ERK proteins belong to the same pathway. This is incorrect. These are two different pathways that cross-talk with each other.

Panel F: which lane correspond to the co-culture with WT versus TNF-Tg B cells? How were cells obtained from the cultures? Were B cells removed from these co-cultures to perform the western blot specifically using MPC?

-We thank the reviewer for pointing out the lack of clarity in the manuscript. We have now corrected this in the revised manuscript with clarifications further summarized below.

Conditioned medium containing LPS+Anti-CD40+IL4 in the absence of B cells had no effect on MSC and OB differentiation as now shown in fig S7B. BM B cells from WT or TNF-Tg mice were purified, and stimulated (S) with 2.5ug/ml Anti-CD40+10ng/ml IL4+10ug/ml LPS or vehicle (U) for 4 hours. CCL3 and TNF protein expression were detected in the culture medium by ELISA, as shown in Fig. 5D&E.

Panel B: B cells used in co-cultures were not activated. We have also clarified that AKT and ERK are two different pathways that cross-talk with each other.

Panel F: We have added the essential labels for each lane. At the end of culture period, B cells were removed and MPCs were specifically analyzed by Western blot. This information has been added to the revised Figure legends.

- 4) *Figure 4: It is not clear whether the B cells were cultivated or not prior to the co-culture with MPC. Panel A: What does n=4 mean? Do the data show a compilation of 4 independent experiments or a representative experiment? This should be better explained also for the other figure panels. The authors show that adding anti-CCL3 or anti-TNF antibodies to the culture increases the ALP area. The effects are however incomplete in both cases. Would neutralization of both factors lead to a complete abrogation of the effect of the B cells? I could not find the references of the reagents used to neutralize CCL3 or TNF in the Materials and Methods. This should be completed. The data shown in Supplementary Fig 3 on the effect of CCL3 on MPC cultures, and on the expression of CCL3 by B cells in situ are interesting. It might be relevant to move some of these data to main figures. It would be useful that the authors perform a different staining to confirm that B220+ cells are indeed B*

cells. B220 is not a strict B cell-specific marker. Staining in addition for immunoglobulin (eg Igk) would provide further strength to these data.

-B cells were not cultured or stimulated before co-culturing with MPCs. Figure legends have been revised for added clarification: experiments were repeated 3-5 times with a representative experiment shown. The question of whether double blockade using anti-CCL3 and anti-TNF antibodies completely suppresses the B cell OB inhibitory effect is an important question. We have now added this experiment (Fig. 4A-C). We have also included the information regarding the neutralizing anti-CCL3 and anti-TNF antibodies in the Materials and Methods. We appreciate

Flow cytometry analysis of SBM cells from a TNF-Tg mouse reveals that the B220+ cells are indeed CD19+ B cells. SBM cells were isolated as noted in the previous figures.

demonstrated that the B220+ cells were largely CD19+, a very specific B cell marker (see left).

the reviewer's comment that the CCL3 effects on MPC cultures and in situ data are interesting and essential and have now moved this CCL3 data from Fig S3 to Fig. 5A-C. Although B220 staining may not be strictly restricted to the B cell compartment, when we isolated cells from the synovium/subchondral bone for RNA sequencing and flow cytometry phenotyping we

5) Figure 5: panel B: the authors should explain how they made the quantification of the osteoblasts. The distribution of the cells is not uniform. Do the data actually show number of OB per mm, or per mm²? Taken together, the data shown in Fig 4 and 5 suggest that B cells control OB formation through production of CCL3 and TNF. However, the effect of the 8-weeks B cell-depletion treatment could affect OB via many different ways others than this one. Thus, the data do not demonstrate directly that B cell production of CCL3 or TNF affects OB in vivo. It is critical that the authors demonstrate this using a direct approach, especially since direct contact between OB and B cells was not obvious in histology. This should be done using mixed BM chimera in which only B cells cannot produce CCL3 or TNF.

-We agree with the reviewer's comment that the distribution of the OBs is not uniform and have reanalyzed the OCN IF with OCN⁺ area /Tissue area (%).

-To demonstrate directly that B cell production of CCL3 or TNF affects OBs in vivo, we have now performed ectopic bone formation experiments using WT MPCs and BM B cells from WT, TNF-Tg, TNF-Tg/CCL3-KO, TNF-Tg/TNF-KO or TNF-Tg/CCL3-KO/TNF-KO mice, as shown in

revised Fig. 5F-H.

6) Figure 6: The reagents used to activate B cells should be better described, with indication of company name and catalog number. The name of the clones should be provided for the antibodies. Were B cells from HD and RA patients used in panel D activated prior to their use?

In supp figure 6, the authors claim that memory B cells are the most relevant B cell subset based on their CD27 expression. However, antibody secreting cells also express high levels of CD27. How can the authors distinguish between the involvement of memory versus antibody-secreting cells?

-The revised Methods now include these critical details. In Panel D, B cells were not activated but directly co-cultured with human MSCs immediately after isolation. In Fig S6, the CD27+CD19+ B cells isolated from peripheral blood (PB) are memory B cells not PCs because that is the dominant CD27 expressing B cell in PB. PCs are extremely sparse in PB as shown in the representative example of flow cytometry.

Flow cytometry of peripheral blood B cells from an RA patient. The plots show that 24.6 % of B cells are CD27+ (including CD27hi plasma cells). However, only a small fraction of these CD27+ B cells are plasma cells, defined as CD19+IgD-CD27hiCD38hi (0.6% of B cells).

REVIEWERS' COMMENTS:

Reviewer #1 (Remarks to the Author):

The authors have sufficiently addressed all of my points.

Reviewer #2 (Remarks to the Author):

The authors have fully addressed my questions.